# Evolutionary unique *N*-glycan-dependent protein quality control system plays pivotal roles in cellular fitness and extracellular vesicle transport in *Cryptococcus neoformans*

Catia Mota[1], Kiseung Kim[1], Ye Ji Son[1], Eun Jung Thak[1], Su-Bin Lee[1], Ju-El Kim[2], Jeong-Kee Yoon[2], Min-Ho Kang[3,4], Heeyoun Hwang[5], Yong-Sun Bahn[6], J Andrew Alspaugh[7], Hyun Ah Kang[1]*

[1]Department of Life Science, Chung-Ang University, Seoul, Republic of Korea; [2]Department of Systems Biotechnology, Chung-Ang University, Gyeonggi-Do, Republic of Korea; [3]Department of Biomedical-Chemical Engineering, The Catholic University of Korea, Bucheon-si, Republic of Korea; [4]Department of Biotechnology, The Catholic University of Korea, Bucheon-si, Republic of Korea; [5]Digital OMICs Research Center, Korea Basic Science Institute, Cheongju-si, Republic of Korea; [6]Department of Biotechnology, Yonsei University, Seoul, Republic of Korea; [7]Department of Medicine, Duke University, Durham, United States

*For correspondence: hyunkang@cau.ac.kr

## eLife Assessment

This **important** study confirms the molecular function of putative components of the N-glycan-dependent endoplasmic reticulum protein quality control (ERQC) system in the pathogen *Cryptococcus neoformans*. The study demonstrates an involvement in fitness, virulence, and the secretion and composition of extracellular vesicles, albeit in ways that are not yet fully understood. The evidence provided is **convincing**, with rigorous, well-controlled assays and the use of complemented strains.

**Abstract** A conserved *N*-glycan-dependent endoplasmic reticulum protein quality control (ERQC) system has evolved in eukaryotes to ensure accuracy during glycoprotein folding. The human pathogen *Cryptococcus neoformans* possesses a unique *N*-glycosylation pathway that affects microbial physiology and interactions with the infected host. To investigate the molecular features and functions of the ERQC system in *C. neoformans*, we characterized a set of mutants with deletion of genes coding for the ERQC sensor UDP-glucose:glycoprotein glucosyltransferase (*UGG1*) and putative α1,2-mannose-trimming enzymes (*MNS1*, *MNS101*, *MNL1*, and *MNL2*). The *ugg1Δ*, *mns1Δ*, *mns101Δ*, and *mns1Δ101Δ* mutants showed alterations in *N*-glycan profiles, defective cell surface organization, decreased survival in host cells, and varying degrees of reduced *in vivo* virulence. The *ugg1Δ* strain exhibited severely impaired extracellular secretion of capsular polysaccharides and virulence-related enzymes. Comparative transcriptome analysis showed the upregulation of protein folding, proteolysis, and cell wall remodeling genes, indicative of induced endoplasmic reticulum stress. However, no apparent changes were observed in the expression of genes involved in protein secretion or capsule biosynthesis. Additionally, extracellular vesicle (EV) analysis combined with proteomic analysis showed significant alterations in the number, size distribution, and cargo

composition of EVs in *ugg1Δ*. These findings highlight the essential role of the functional ERQC system for cellular fitness under adverse conditions and proper EV-mediated transport of virulence factors, which are crucial for the full fungal pathogenicity of *C. neoformans*.

## Introduction

Glycoproteins, destined for the secretory pathways, enter the endoplasmic reticulum (ER) lumen where protein folding occurs (**Rapoport, 2007**). The accumulation of misfolded proteins affects cell viability and homeostasis; therefore, eukaryotes evolved a conserved ER quality control (ERQC) system that recognizes folding defects, repairs them, or ensures the translocation of irreparable misfolded proteins into the cytosol for proteasome-mediated degradation via the ER-associated degradation (ERAD) system (**Thibault and Ng, 2012**; **Xu and Ng, 2015**; **Balchin et al., 2016**), a process that heavily relies on *N*-glycosylation to determine protein folding conformation (**Aebi, 2013**; **Varki, 2017**).

Many aspects of *N*-glycosylation are highly conserved, and most eukaryotes initiate this process by synthesizing a Dol-PP-linked $Glc_3Man_9GlcNAc_2$ oligosaccharide as a common core *N*-glycan, which attaches to nascent polypeptides in the ER via an asparagine residue. Glucosidases I and II (Gls1 and Gls2) remove two glucose residues from the core oligosaccharide. Proteins containing monoglucosylated *N*-glycans ($Glc_1Man_9GlcNAc_2$) enter a calnexin and/or calreticulin (CNX/CRT in mammals; Cne in yeast) chaperone-mediated folding cycle. Finally, Gls2 cleaves the remaining glucose residue. If proteins are misfolded, they are recognized by the ERQC checkpoint enzyme UDP-glucose:glycoprotein glucosyl-transferase (UGGT), which reglucosylates them for re-entry into the folding cycle (**Figure 1A**). Then, the *N*-glycans of the accurately folded proteins are further processed by Gls2 and α1,2-mannosidase I (Mns1) and moved to the Golgi apparatus. However, irreparably misfolded glycoproteins are targeted for ERAD, where they undergo demannosylation and retro-translocation for proteasomal degradation in the cytosol. Recently, *N*-glycan precursors in the ER of some protists and fungi were found to be shorter than the typical 14-sugar *N*-glycan precursors in most eukaryote organisms. The length of these *N*-glycan precursors significantly impacts *N*-glycan-dependent QC of glycoprotein folding and ERAD (**Banerjee et al., 2007**; **Samuelson and Robbins, 2015**).

The basidiomycetous fungus *Cryptococcus neoformans* is an opportunistic encapsulated human pathogen that primarily affects immunocompromised individuals, causing fatal meningoencephalitis (**Gottfredsson and Perfect, 2000**; **Kwon-Chung et al., 2000**). The *N*-glycosylation pathway of *C. neoformans* is evolutionarily conserved; nevertheless, the structure and biosynthesis of its *N*-glycans include several unique features (**Park et al., 2012**). *C. neoformans* contains serotype-specific high-mannose-type *N*-glycans with or without a β–1,2-xylose residue attached to the trimannosyl core. Additionally, acidic *N*-glycans of *C. neoformans* contain xylose phosphates attached to the mannose residues both within the *N*-glycan core and outer mannose chains. The intact core *N*-glycan structure is crucial for *C. neoformans* pathogenicity (**Thak et al., 2020**); hence, alterations in the *N*-glycan structure modulate the interaction between the cell surface mannoproteins and host cells (**Lee et al., 2023**). Additionally, *C. neoformans* lacks homologous genes to the Asn-Linked Glycosylation (ALG) genes *ALG6*, *ALG8*, and *ALG10*, which are evolutionary conserved across most eukaryotic organisms and encode the glucosyltransferases that add the glucose residues to the core *N*-glycan before its attachment to proteins (**Park et al., 2012**). $Man_7GlcNAc_2$ and $Man_8GlcNAc_2$ without glucose residues are primarily detected in Dol-PP-linked glycans of *C. neoformans* (**Samuelson et al., 2005**). Moreover, the mature core *N*-glycan structures assembled on the cell surface mannoproteins of *C. neoformans* are primarily $Man_{6-7}GlcNAc_2$, which are shorter than the expected $Man_8GlcNac_2$ (**Park et al., 2012**). This observation led to the speculation that the terminal α–1,2-mannose residues of *C. neoformans* *N*-linked glycans may be more susceptible to trimming by ER α–1,2 mannosidases due to the lack of glucose residues compared to most eukaryotes. Alternatively, the presence of multiple α–1,2 manno-sidases may generate more extensively trimmed core *N*-glycans in *C. neoformans*.

*C. neoformans* employs several virulence factors, including an extensive polysaccharide capsule composed of glucuronoxylomannan (GXM) and galactoxylomannan (GalXM) (**Doering, 2009**), melanin (**Qiu et al., 2012**), and various extracellular enzymes such as phosphatase and urease (**Singh et al., 2013**), all of which contribute to immune evasion and enhance fungal pathogenicity. To facilitate the export of these virulence-associated molecules, *C. neoformans* employs both conventional (**Yoneda and Doering, 2006**; **Panepinto et al., 2009**) and unconventional secretion pathways (**Rodrigues**

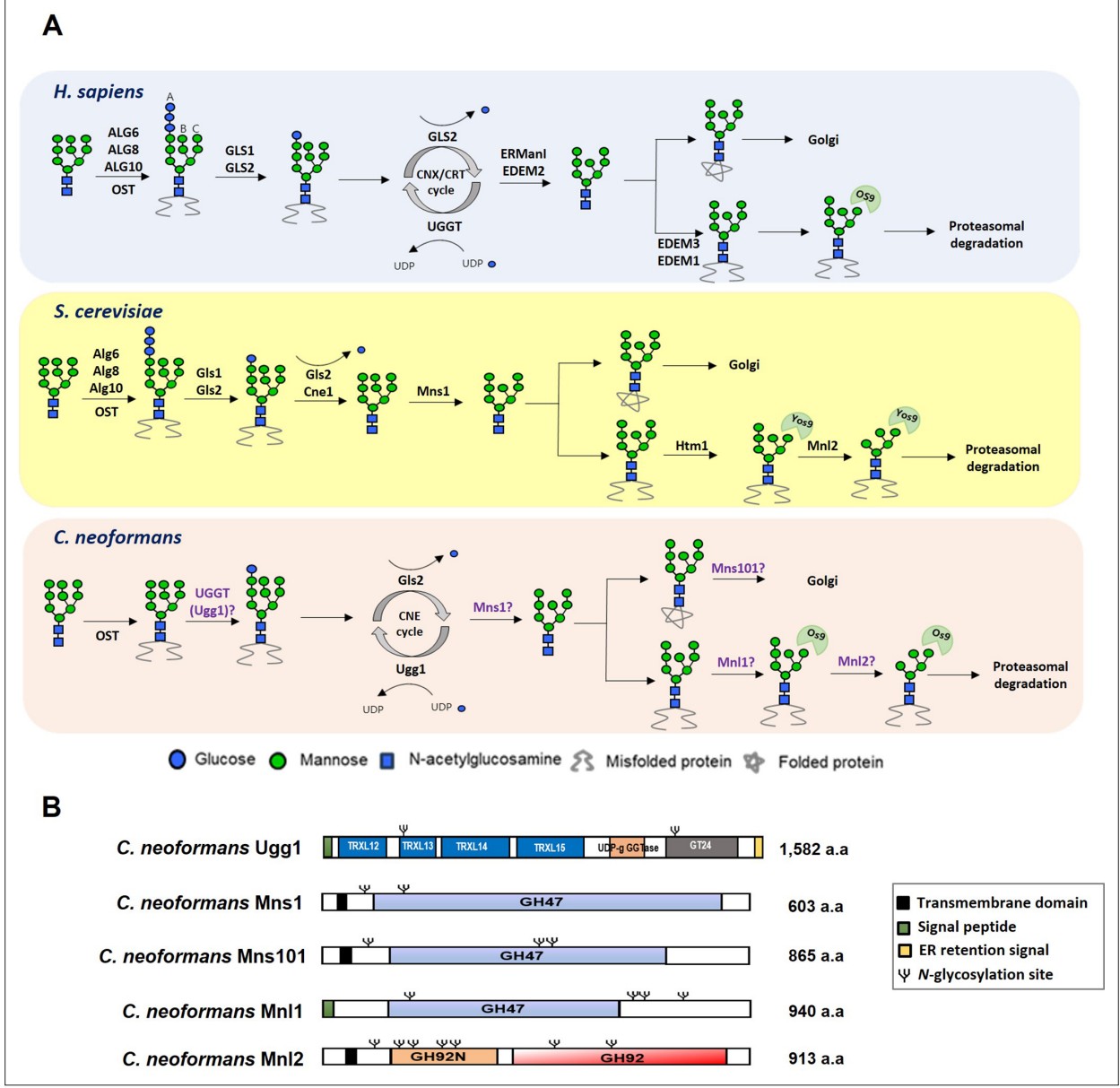

**Figure 1.** Presence of UDP-glucose:glycoprotein glucosyltransferase (UGGT) and α1,2-mannosidases as endoplasmic reticulum protein quality control (ERQC) components in *Cryptococcus neoformans*. (**A**) Schematic representation of the ERQC pathway in *Homo sapiens*, *Saccharomyces cerevisiae*, and *Cryptococcus neoformans*. In mammals, the oligosaccharyltransferase (OST) complex attaches $Glc_3Man_9GlcNAc_2$ to nascent polypeptides, followed by glucose trimming by glucosidases I and II (GLS1/GLS2). This generates $Glc_1Man_9GlcNAc_2$, which binds to calnexin (CNX) and calreticulin (CRT) for folding. UGGT reglucosylates misfolded proteins, allowing refolding, while properly folded proteins undergo mannose trimming by ERManI before Golgi transport. In contrast, fungal ERQC systems differ in key components. *S. cerevisiae* lacks UGGT, relying instead on Gls1/Gls2 and calnexin (Cne1). *C. neoformans* possesses UGGT but lacks ER glucosyltransferases (Alg6, Alg8, and Alg19) and CRT, resulting in a distinct ERQC system. (**B**) Domain structures of proteins encoded by *C. neoformans UGG1* (CNAG_03648), *MNS1* (CNAG_02081), *MNS101* (CNAG_03240), *MNL1* (CNAG_01987), and *MNL2* (CNAG_04498).

The online version of this article includes the following figure supplement(s) for figure 1:

**Figure supplement 1.** Evolutionary conservation and divergence of endoplasmic reticulum quality control (ERQC)-related gene homologs in eukaryotes.

**Figure supplement 2.** Domain analysis of putative *C. neoformans* endoplasmic reticulum quality control (ERQC)-associated proteins.

*et al., 2007*; *Casadevall et al., 2019*; *Rizzo et al., 2021*). Classical secretion relies on the ER-Golgi network, while non-classical mechanisms, such as extracellular vesicle (EV)-mediated transport, provide an alternative route for cargo delivery beyond the conventional secretory pathway. *C. neoformans* EVs have been described as a heterogeneous population of 'virulence bags' containing numerous fungal survival and pathogenicity-associated cargo. Proteomic analyses of EV cargo have identified several cell surface glycoproteins, including members of the CDA family, well-known immunomodulators (*Specht et al., 2017*; *Rizzo et al., 2021*). The ERQC system is essential for maintaining protein homeostasis by ensuring proper folding, glycosylation, and directed trafficking of secreted proteins to downstream compartments within the conventional secretory pathway. Defects in ERQC affect the fidelity of protein secretion not only by accumulation or degradation of misfolded proteins, leading to the impaired secretion of functional proteins, but also by the escape of misfolded forms, thus resulting in increased secretion of non-properly processed proteins (*Marcus and Perlmutter, 2000*; *Chen et al., 2024*). However, the potential of ERQC to modulate EV-mediated protein trafficking is not yet systematically investigated.

This study presents the first systematic analysis of the *N*-glycan-dependent ERQC in *C. neoformans* using mutant strains lacking ERQC gene homologs. Our findings highlight the critical roles of ERQC in maintaining cellular fitness and facilitating EV-mediated transport of virulence factors.

## Results

### Evolutionary unique features of *C. neoformans* ERQC components

We performed BLAST analysis of the *C. neoformans* H99 genome to identify the ERQC pathway-associated homologous genes of *C. neoformans*, followed by comparison with the ERQC components in other eukaryotic organisms (*Figure 1—figure supplement 1A*). Unlike most eukaryotes, several yeast species within the Ascomycota phylum, such as *Saccharomyces cerevisiae* and *Candida albicans,* do not possess a functional UGGT. In contrast, UGGT homologs were identified in most Basidiomycota fungal species, including *C. neoformans* (*Figure 1—figure supplement 1A and B*). The *C. neoformans* UGGT (CNAG_03648), which has been named as Ugg1, consists of 1,582 amino acids (aa) and features a signal peptide (1–20 aa) along with four tandem-like thioredoxin-like (TRXL) domains: TRXL12 (33–320 aa), TRXL13 (28–416 aa), TRXL14 (432–618 aa), and TRXL15 (712–950 aa). Ugg1 also contains a glucosyltransferase (GT) 24 domain with a DXD motif (1,369–1,371 aa), and a KDEL-like ER retention signal (1,579–1,582 aa), which facilitates its retrieval from the Golgi apparatus (*Figure 1B*, *Figure 1—figure supplement 1A*) .

Additionally, we identified two *C. neoformans* ORFs, Mns1 (CNAG_02081) and Mns101 (CNAG_03240), as homologs of the eukaryote α1,2-mannosidase I, which processes *N*-glycans before exporting them to the Golgi. The *C. neoformans* Mns1 and Mns101 show 42.4% and 28.8% amino acid identities to *S. cerevisiae* ER α1,2-mannosidase I, respectively, and share 30.9% identity between them. Furthermore, *C. neoformans* Mnl1 (CNAG_01987) and Mnl2 (CNAG_04498) were identified as putative components of ERQC in *C. neoformans* (*Figure 1—figure supplement 1A and C*). *C. neoformans* Mnl1 is a homolog of the yeast α1,2-mannosidase-like protein Htm1, which processes *N*-glycans targeted for ERAD. In contrast, *C. neoformans* Mnl2 encodes a mannosidase that does not share significant similarity with mannosidases from other eukaryotes. Mns1, Mns101, and Mnl1 possess glucosyl hydrolase (GH) 47 domains, essential for mannosidase activity, whereas Mnl2 contains a GH92 domain, also associated with mannosidase activity (*Figure 1B*). The Mns1 and Mnl1 families are characterized by conserved cysteine (Cys) and alanine (Ala) residues within their activity domains, as previously described (*Jakob et al., 2001*). These conserved residues are present in *C. neoformans* Mns1, Mns101, and Mnl1 but are absent in Mnl2 (*Figure 1—figure supplement 2B*). Notably, Mns101 and Mnl2 are Basidiomycota-specific proteins (*Figure 1—figure supplement 1A*) and appear to have diverged early from other fungal mannosidase clades based on phylogenetic analysis (*Figure 1—figure supplement 1C*). Interestingly, neither Mnl1 nor Mnl2 contains canonical KDEL/HDEL-like ER retention signals. In *S. cerevisiae*, the ER retention of Mnl1/Htm1 is mediated through its interaction with protein disulfide isomerase Pdi1, which carries an HDEL sequence (*Gauss et al., 2011*). Similarly, *C. neoformans* Mnl1 and Mnl2 may employ a non-canonical retention mechanism, likely facilitated by interactions with other ER-resident proteins, to achieve ER localization.

## Loss of *UGG1*, *MNS1*, and *MNS101* causes alteration of *N*-glycan profiles in *C. neoformans*

We performed high-performance liquid chromatography (HPLC) analysis of the cell wall mannoproteins (cwMPs) from both the wild-type (WT) and ERQC mutant strains to investigate the ERQC malfunction-induced structural differences in the *N*-glycosylation profile (*Figure 2*). The HPLC profiles of cwMPs from the WT strain showed an M8 peak as the major species. This peak corresponded to *N*-glycans with eight mannose residues (*Figure 2A*, top). The glycan structure at the M8 peak (Man$_8$GlcNAc$_2$) in the WT primarily corresponded to the Man$_7$GlcNAc$_2$ core *N*-glycan with an additional mannose residue that is linked via an α1,6-linkage and added in the Golgi apparatus (*Park et al., 2012*). In the *ugg1Δ* mutant, M8 was also the main *N*-glycan species, but the pools of hypermannosylated *N*-glycans (larger than the M11 peak) were markedly reduced (*Figure 2A*, middle). The altered *N*-glycan profile of the *ugg1Δ* mutant was restored to that of the WT after complementation with the WT *UGG1* gene (*Figure 2A*, bottom). The lectin blotting analysis using *Galanthus nivalis* agglutinin (GNA), which specifically binds to terminal α1,2-, α1,3-, and α1,6-linked mannose residues, showed a distinctive increase of glycoproteins with lower molecular weight in the secretory proteins in *ugg1Δ* compared to those in WT (*Figure 2B*). This observation aligns with the decrease of hypermannosylated *N*-glycans in the *ugg1Δ* mutant. Overall, these results suggest that Ugg1 is involved in mediating the hypermannosylation of *N*-glycans in the Golgi apparatus.

We also assessed putative α1,2-mannosidase I genes for roles in protein mannosylation. The *mns1Δ* mutant *N*-glycan profile showed a peak shift from an M8 to an M9 form, which strongly indicates that *C. neoformans* Mns1 could be the primary ER α1,2-mannosidase I (*Figure 2C*, *mns1Δ*). Notably, the loss of Mns101, which is present only in Basidiomycota, increased the fractions containing hypermannosylated glycans (>M10) while maintaining M8 as a primary core *N*-glycan form. This suggests that the basidiomycete-specific Mns101 may potentially be a novel α1,2-mannosidase that functions to remove mannose residues from hypermannosylated *N*-glycans in the Golgi apparatus or further trims the M8 glycan in the ER before the glycoproteins are transported to the Golgi (*Figure 2C*, *mns101Δ*). The matrix-assisted laser desorption ionization-time of flight (MALDI-TOF) mass spectrometry analysis of neutral *N*-glycans (*Figure 2C*) further confirmed the shift of the major M8 peak in the WT strain to M9 in the *mns1Δ* mutant and the increase in the hypermannosylation profile in the *mns101Δ* mutant (*Figure 2D*). The *N*-glycan profiles of the *mns1Δ101Δ* double mutant showed the combined effect of each null mutation, and it showed both increased hypermannosylated glycans and a shift from the M8 to the M9 peak in both the HPLC (*Figure 2C*, *mns1Δ101Δ*) and MALDI-TOF analyses. This indicates that Mns1 and Mns101 serve as mannosidases and play independent roles at different stages of *N*-glycan processing in *C. neoformans*.

In contrast, the loss of both *MNL1* and *MNL2* did not show notable differences in the HPLC profile of *N*-glycans from cell surface mannoproteins (*Figure 3—figure supplement 1A*). Considering the expected function of Mnl1 and Mnl2 in ERAD, their substrates are likely misfolded proteins that have not been transported to the Golgi apparatus. Thus, we speculate that only the normally processed *N*-glycan profiles of cell surface mannoproteins in the *mnl1Δ mnl2Δ* mutant strain were observed.

## Loss of ERQC components results in defective growth fitness and increased stress sensitivity

To investigate the changes in ERQC-related gene expression under stress conditions, we conducted quantitative reverse transcription PCR (qRT-PCR) analysis of *C. neoformans* following treatment with tunicamycin (TM, 5 µg/ml), a well-established inhibitor of *N*-glycosylation, or dithiothreitol (DTT, 20 mM), which disrupts disulfide bond formation and induces the accumulation of misfolded proteins. Additionally, we examined gene expression after growth at 37 °C (*Figure 3A*). When misfolded proteins accumulate in the ER, the unfolded protein response (UPR) system is activated to restore homeostasis (*Chakraborty et al., 2016*).

As expected, *KAR2*, encoding a molecular chaperone associated with the UPR system, was upregulated in response to TM treatment, DTT treatment, and heat stress at 37 °C. The ERQC components *UGG1*, *MNS1*, *MNS101*, *MNL1*, and *MNL2* were upregulated by DTT treatment; however, only *UGG1* was slightly induced following TM treatment. The high-temperature conditions also induced the expression of *KAR2*, *UGG1*, and *MNL2*. Notably, DTT treatment induced a 10-fold higher expression of *MNS1* and *MNL1*. These results strongly suggest that the *C. neoformans* genes *UGG1*, *MNS1*,

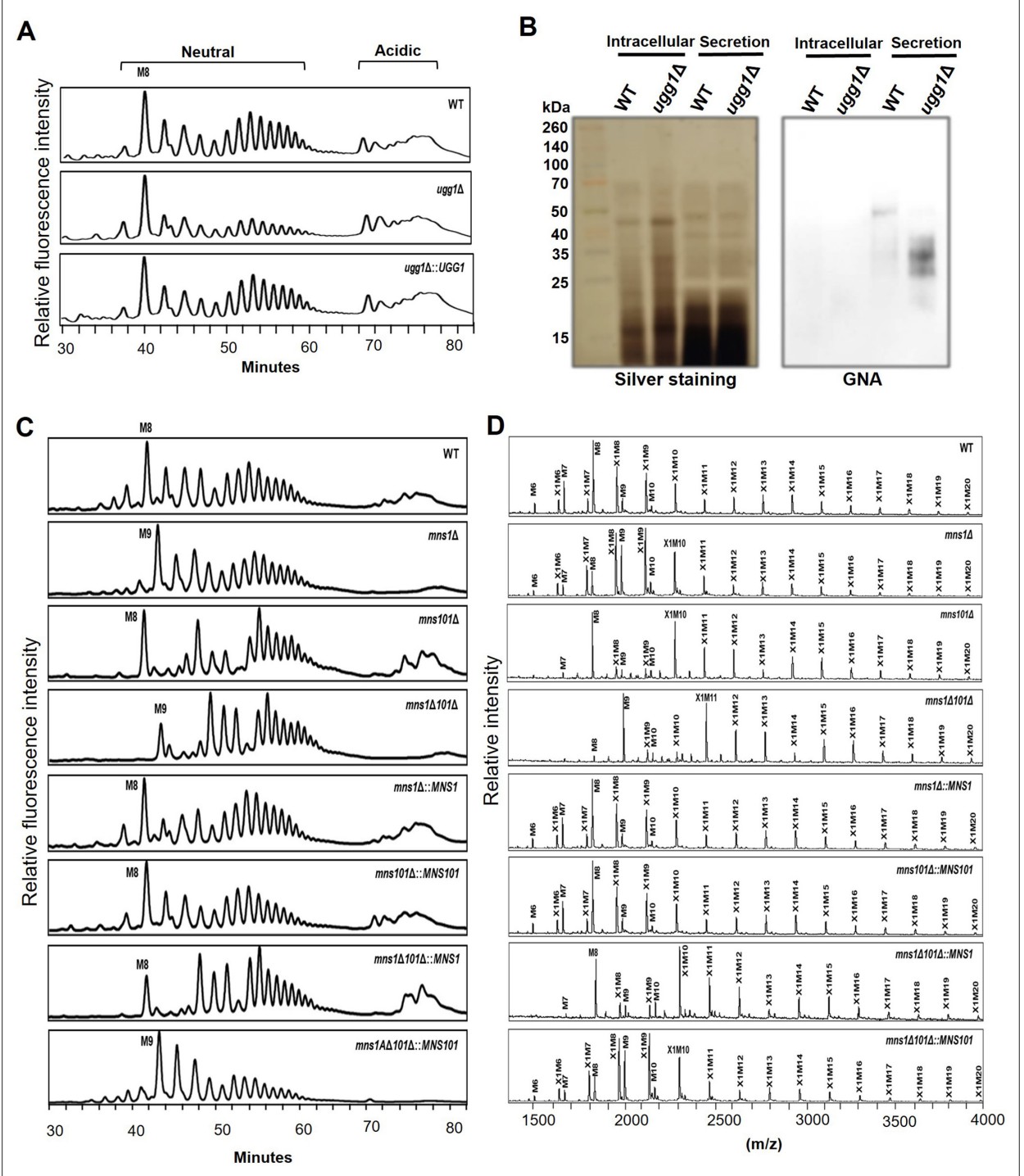

**Figure 2.** *N*-glycan profile analysis of *C. neoformans* endoplasmic reticulum quality control (ERQC) mutant strains. (**A**) High-performance liquid chromatography (HPLC)-based analysis of *N*-glycan profiles of the *ugg1Δ* mutant. (**B**) Lectin blotting of sodium dodecyl sulfate (SDS)-polyacrylamide gels containing intracellular or proteins secreted into the culture supernatants of the wild type (WT) and *ugg1Δ* strains. Yeast cells were cultivated in YPD medium for 24 h, harvested, and subjected to sample preparation of soluble intracellular proteins and secreted proteins. The proteins (30 μg) were loaded on 15% SDS-polyacrylamide gel and analyzed using silver staining (left) or blotting (right) with *Galanthus nivalis* agglutinin conjugated to horseradish peroxidase (GNA-HRP, Roche). (**C**) HPLC analysis of total *N*-glycan profiles of *mns1Δ*, *mns101Δ*, and *mns1Δ101Δ* mutants. (**D**) Matrix-assisted laser desorption ionization-time of flight (MALDI-TOF) profiles of neutral *N*-glycans of *mns1Δ*, *mns101Δ*, and *mns1Δ101Δ* mutants. The *N*-glycans of cell wall mannoproteins from *C. neoformans* cells were AA-labeled and analyzed using HPLC. For MALDI-TOF analysis, neutral *N*-glycan fractions were obtained from the HPLC fractionation of total *N*-glycans.

*Figure 2 continued on next page*

*Figure 2 continued*

The online version of this article includes the following source data for figure 2:

**Source data 1.** Original gel displayed in *Figure 2B*, indicating the relevant strains.

**Source data 2.** Uncropped gel displayed in *Figure 2B*.

**Source data 3.** Original blot membrane displayed in *Figure 2B*, indicating the relevant strains.

**Source data 4.** Uncropped blot displayed in *Figure 2B*.

*MNS101*, *MNL1*, and *MNL2* are crucial components of the ERQC and ERAD systems, although each respond uniquely to various cell/ER stress conditions.

We next investigated the growth of the *ugg1Δ* mutant under various stress-inducing conditions to elucidate the roles of the ERQC components in ER stress response and adaptation, which are closely associated with *C. neoformans* virulence. The *ugg1Δ* mutant showed impaired growth even under normal growth conditions (*Figure 3B*) and increased sensitivity to ER stress-inducing agents such as DTT, TM, and cell wall stressors, including as calcofluor white (CFW), Congo red (CR), sodium dodecyl sulfate (SDS), and caffeine. Additionally, *ugg1Δ* showed hindered growth in the presence of antifungal drugs such as azole agents (fluconazole and ketoconazole) and the glucose transport inhibitor fludioxonil. These findings suggest that Ugg1 is crucial for the robust growth and survival of *C. neoformans* under various stress conditions. In contrast, the single-deletion (*mns1Δ* and *mns101Δ*) and double-deletion (*mns1Δ101Δ*) strains did not exhibit noticeable phenotypic changes except for a slight increase in sensitivity of *mns101Δ* and *mns1Δ101Δ* to higher fludioxonil concentrations (*Figure 3—figure supplement 1B*). Similarly, neither the single nor double disruption of *MNL1* and *MNL2* produced detectable changes in the tested growth conditions (*Figure 3—figure supplement 1C*).

Amino acid analogues have been used for evaluating the functionality of protein folding as its incorporation into newly synthesized proteins interrupts proper protein folding (*Cowie et al., 1959*; *Trotter et al., 2002*). We hypothesized that, in the context of a defective ERQC pathway, misfolded proteins cannot be adequately repaired and accumulate, thus triggering ER stress, which may ultimately inhibit cell growth in the presence of amino acid analogues. In the presence of the leucine analogue 5',5',5'-trifluoroleucine (TFL), *ugg1Δ* showed noticeably inhibited growth, whereas the WT strain showed no growth inhibition (*Figure 3C*). The *mns1Δ101Δ* strain also exhibited slightly increased sensitivity to TFL compared with that of the WT strain.

As a defense mechanism against misfolded protein accumulation-mediated ER stress, UPR is induced by the unconventional splicing of the *HXL1* transcription factor in *C. neoformans* (*Cheon et al., 2011*). RT-PCR analysis revealed a significantly elevated level of spliced *HXL1* in the *ugg1Δ* strain compared to the WT, even under normal culture conditions (*Figure 3D*), strongly indicating that the loss of *UGG1* leads to the accumulation of misfolded proteins, triggering ER stress under standard growth conditions. Furthermore, green fluorescence protein (GFP)-tagged Ugg1, Mns1, and Mns101 proteins colocalized with the ER marker, supporting their role as confirming them to be functional ERQC components based on their subcellular localization in the ER (*Figure 3—figure supplement 2*).

## ERQC defects lead to virulence attenuation in *C. neoformans*

We investigated the effects of a defective ERQC on *in vitro* virulence phenotypes of *C. neoformans* by analyzing capsule and melanin production as these are two of the major virulence factors of *C. neoformans*. Culturing on L-DOPA-containing plates showed that melanin production was reduced in the *ugg1Δ* strain compared with that in the WT (*Figure 4A*, top), whereas the *mns1Δ101Δ* mutant cells did not show detectable defects at either 30 °C or 37 °C (*Figure 4A*, bottom). Considering the decreased growth of the *ugg1Δ* strain, the melanin production of *C. neoformans* cells cultivated in liquid L-DOPA was measured and normalized by cell density, which further confirmed the decreased melanin production activity of the *ugg1Δ* strain (*Figure 4B*). India ink staining of the capsule showed significantly reduced capsule thickness in the *ugg1Δ* strain and a moderate defect in the *mns1Δ101Δ* strain (*Figure 4C*).

Defects in the ERQC system led to a more apparent decrease in pathogenicity in a murine model of systemic cryptococcosis. Notably, the *ugg1Δ* strain was almost avirulent with no detectable signs of illness in the infected animals (*Figure 4D*). Although *MNS1* and *MNS101* single disruptions did

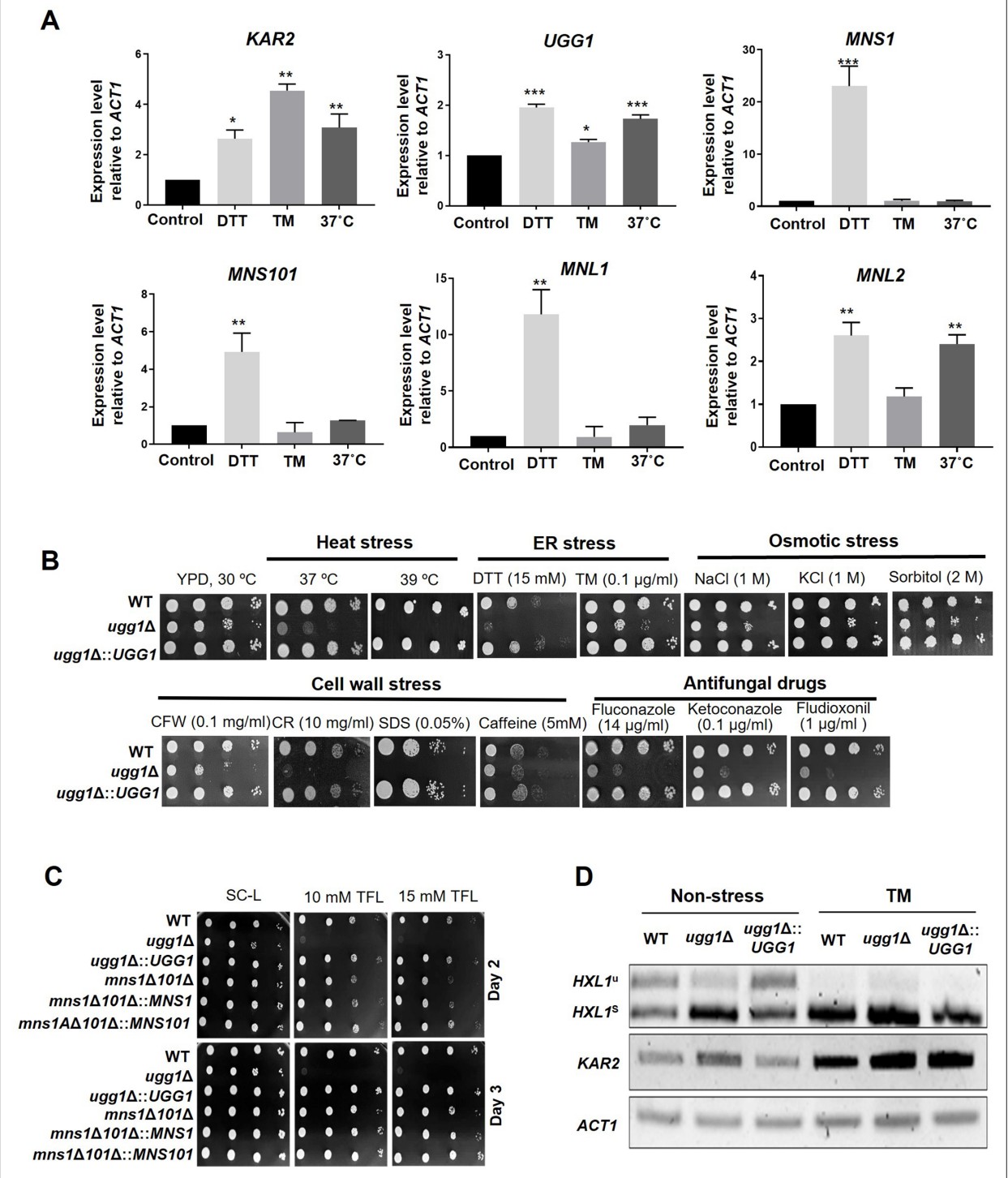

**Figure 3.** Growth phenotype of *C. neoformans* endoplasmic reticulum quality control (ERQC) mutant strains. (**A**) Expression analysis of ERQC genes in *C. neoformans*. Yeast cells were cultured in YPD medium to a mid-logarithmic phase and exposed to dithiothreitol (DTT; 20 mM), tunicamycin (TM; 5 μg/ml), or cultured at 37 °C for 1 h. The relative transcript levels of *C. neoformans* genes were analyzed using qRT-PCR and normalized with that of *ACT1*. Error bars represent standard deviation of duplicated assays. All statistical data were determined based on one-way ANOVA and Dunnett's post hoc test. ***p<0.0005, **p<0.003, ***p<0.005, *p<0.05. (**B**) Spotting analysis of *C. neoformans ugg1Δ* mutant strains under various stress conditions such as heat stress (37 °C and 39 °C), ER stress (DTT and TM), cell-wall stress (CFW: calcofluor white; CR: Congo red; SDS: sodium dodecyl sulfate; caffeine), osmotic stress (NaCl, KCl, sorbitol), and treatment with antifungal drugs (fluconazole, ketoconazole, fludioxonil). (**C**) Growth analysis in the presence of 5',5',5'-trifluoroleucine (TFL). Respective strains were spotted on SC-Leucine media with or without TFL supplementation. Plates were incubated for 3 days at 30 °C. (**D**) RT-PCR analysis of *IRE1*-dependent splicing of *HXL1*. Strains were cultured in YPD supplemented with 5 μg/ml TM.

*Figure 3 continued on next page*

*Figure 3 continued*

The online version of this article includes the following source data and figure supplement(s) for figure 3:

**Source data 1.** Raw data related to *Figure 3*.

**Source data 2.** Original gel displayed in *Figure 3D* (upper panel), indicating the relevant strains.

**Source data 3.** Uncropped gels displayed in *Figure 3D* (upper panel).

**Source data 4.** Original gel displayed in *Figure 3D* (middle panel), indicating the relevant strains.

**Source data 5.** Uncropped gel displayed in *Figure 3D* (middle panel).

**Source data 6.** Original gel displayed in *Figure 3D* (lower panel), indicating the relevant strains.

**Source data 7.** Uncropped gel displayed in *Figure 3D* (lower panel).

**Figure supplement 1.** Phenotype characterization of *C. neoformans* α1,2-mannosidases.

**Figure supplement 2.** Subcellular localization of GFP-tagged Ugg1, Mns1, and Mns101.

not cause a detectable decrease in *C. neoformans* pathogenicity, their double disruption resulted in decreased virulence. Histopathological analysis of infected mice lungs suggested significantly reduced lung colonization by both *ugg1Δ* and *mns1Δ101Δ* cells (*Figure 4—figure supplement 1A*). Additionally, the fungal burden of *ugg1Δ*-infected animals showed a notable decrease in organ colonization at 60 days post-infection (dpi) (*Figure 4—figure supplement 1B*). Analysis of the fungal burden at 7 dpi showed significantly reduced organ colonization of the *mns1Δ101Δ* mutant compared with that of the WT (*Figure 4—figure supplement 1C*). Consistent with the *in vivo* survival data, the number of surviving *ugg1Δ* and *mns1Δ101Δ* cells within the macrophage-like cell line J447A.1 was significantly lower compared with that of the WT strain (*Figure 4E*). These data suggest that a functional ERQC system is critical for maintaining full *in vivo* pathogenicity and ensuring robust survival within host immune cells.

## Loss of Ugg1 function manifests defects in extracellular transport of GXM

Cryptococcal capsule materials are synthesized intracellularly and secreted to the cell surface. Subsequently, they are assembled and bound to the cell wall (*Yoneda and Doering, 2006*). The observed capsule defects in the *ugg1Δ* and *mns1Δ101Δ* mutant strains may be attributed to issues in the capsule polysaccharide synthesis or transport steps. Alternatively, defective capsule formation could arise from an increased shedding of capsule polymers, preventing their proper attachment to the cell surface. To determine the cause of the capsule defects, we examined the amounts of capsular polysaccharides synthesized intracellularly and those shed into the culture supernatant by a capsule blotting assay, using the 18B7 antibody directed against GXM, the major capsule component (*Casadevall et al., 1998*). As controls, we also tested capsule production and secretion in the *cap59Δ* strain with, which has a defect in GXM synthesis (*Grijpstra et al., 2009*), and the *rim101Δ* strain, having a capsule attachment defect despite normal polysaccharide synthesis and secretion (*O'Meara et al., 2010*). Intracellular GXM quantities did not notably differ between the WT, *ugg1Δ*, *mns1Δ101Δ*, and *rim101Δ* strains, although *ugg1Δ* showed slightly reduced GXM level (*Figure 5A*, left). In contrast, no intracellular GXM was detected in *cap59Δ*, consistent with its defect in GXM synthesis. Moreover, the *ugg1Δ* strain was defective in capsule polymer secretion/shedding, whereas the *mns1Δ101Δ* strain displayed enhanced shedding of GXM-containing polymers that appeared to be shorter in length than the WT-type polymers, suggestive of poor polysaccharide polymerization (*Figure 5A*, right). Therefore, these results suggest that *ugg1Δ* contains a defect in polysaccharide trafficking to the extracellular space, whereas *mns1Δ101Δ* has a defect in polysaccharide polymerization, resulting in defective capsule elaboration, albeit with differential degrees of severity in both ERQC mutants.

Furthermore, to determine whether the ERQC mutants are also impaired in capsule attachment, we performed a capsule transfer assay to assess whether exogenously shed capsule polysaccharides can bind to acapsular mutants (*Reese and Doering, 2003*). The hypocapsular *rim101Δ* mutant did not show reassociation of capsular polysaccharides; however, the acapsular *cap59Δ* mutant showed recovery of capsule formation when co-incubated with exogenous GXM (*Figure 5B*). Similarly, *ugg1Δ* reverted its acapsular phenotype on incubation with the WT-shed polysaccharides, strongly suggesting that the defective capsule phenotype of *ugg1Δ* is primarily due to defective GXM secretion rather

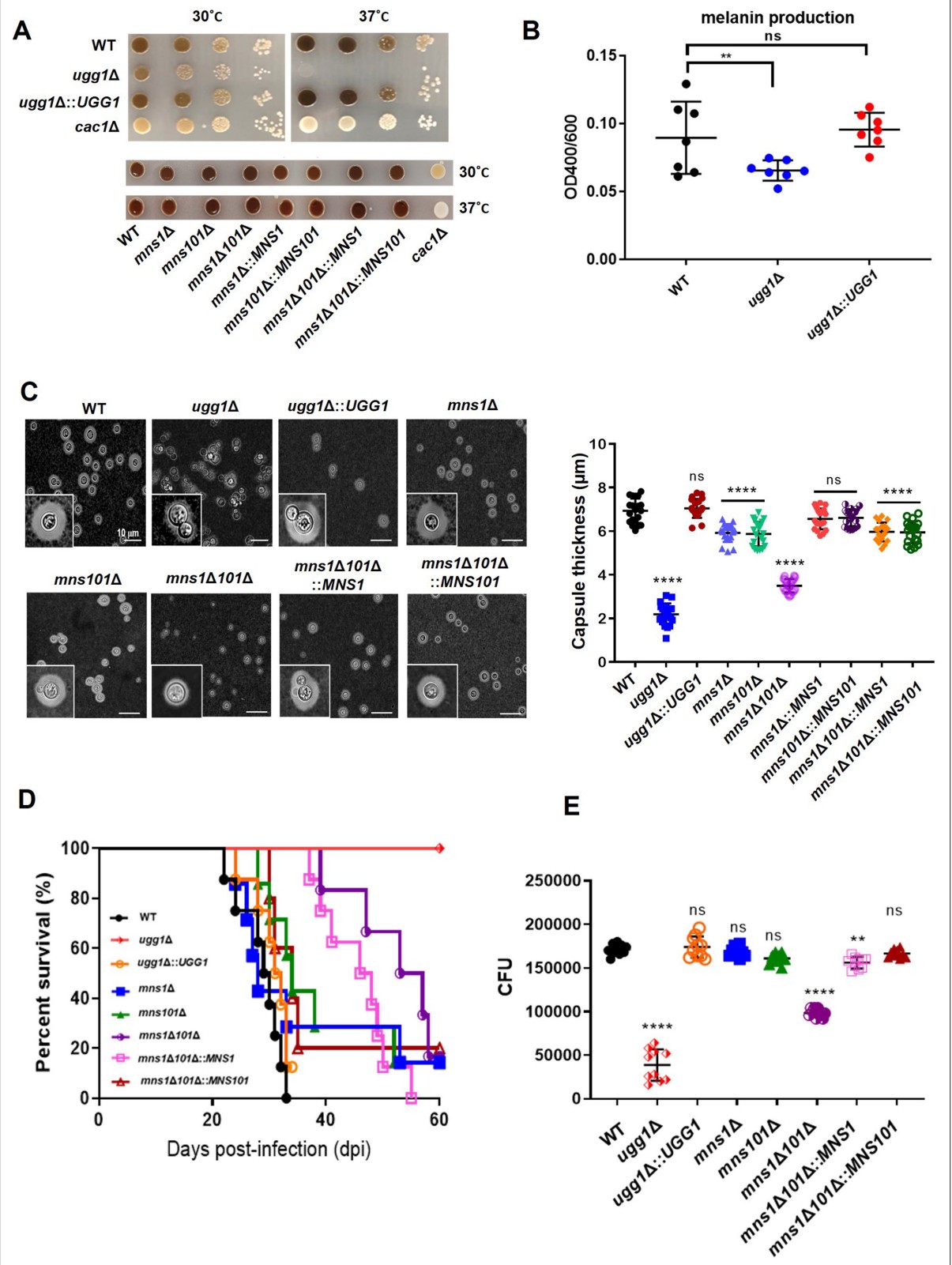

**Figure 4.** *In vitro* and *in vivo* virulence-associated phenotypes of *C. neoformans UGG1, MNS1,* and *MNS101* mutant strains. (**A**) Melanin synthesis analysis on L-DOPA plates. WT, *ugg1Δ, ugg1Δ::UGG1, mns1Δ, mns101Δ, mns1Δ101Δ, mns1Δ101Δ::MNS1, mns1Δ101Δ::MNS101,* and *cac1Δ* (negative control) strains were serially diluted, plated on L-DOPA plates, and incubated at 30 °C and 37 °C. (**B**) Melanin synthesis activity per cell. WT, *ugg1Δ, ugg1Δ::UGG1 w*ere cultured in liquid L-DOPA medium. The amount of melanin in the culture supernatant was measured and normalized by cell density.

*Figure 4 continued on next page*

*Figure 4 continued*

(**C**) Capsule formation. Cells were cultured for 2 days in 10% Sabouraud media at 30 °C and observed under the microscope. Statistical significance: ****p<0.0001, ns, not significant. (**D**) *In vivo* virulence analysis. A/Jcr mice (n = 8) were infected with $10^5$ cells of WT, *ugg1Δ*, and *ugg1Δ::UGG1*, *mns1Δ*, *mns101Δ*, *mns1Δ101Δ*, *mns1Δ101Δ::MNS1*, and *mns1Δ101Δ::MNS101* strains, and survival was monitored for 2 months. (**E**) Survival of *C. neoformans* in macrophages. Survival of *C. neoformans* cells within the J774A.1 macrophage-like cell line was determined by counting colony-forming unit (CFU) obtained from lysed macrophages from two biologically independent experiment sets. ****p<0.0001, ***p<0.0005, *p<0.05, ns, not significant. All statistical data were determined based on one-way ANOVA and Dunnett's post hoc test.

The online version of this article includes the following source data and figure supplement(s) for figure 4:

**Source data 1.** Raw data related to *Figure 4*.

**Figure supplement 1.** Fungal burden of mice infected with *C. neoformans* endoplasmic reticulum quality control (ERQC)-related mutant cells.

than impaired attachment of capsule polymers. We further conducted a capsule transfer assay using capsular polysaccharides shed from *mns1Δ101Δ*. The *cap59Δ* mutant reverted the acapsular phenotype by attaching the *mns1Δ101Δ*-shed polymers. However, the fluorescence intensity of the capsule generated was significantly lower than that of the cells attached with polysaccharides obtained from the WT. Although the decreased length of capsule polysaccharides should be validated by techniques specifically measuring GXM size (*De Jesus et al., 2010*), this result suggests that the *mns1Δ101Δ* double mutant secretes incomplete capsule polysaccharides (*Figure 5C*), leading to a hypocapsular phenotype. These results collectively indicate that disrupting the ERQC results in impaired capsule formation by defective GXM trafficking to the extracellular space.

Transmission electron microscopy (TEM) imaging of the WT and mutant cells distinctly showed a diminished capsule structure and loss of capsule shedding in the *ugg1Δ* strain compared with that of the WT strain (*Figure 5D*). Additionally, we observed considerable thinning of the cell walls in both ERQC mutant cells. Notable changes in intracellular structures along with an increase in the number of pigmented vesicles were observed in the mutant cells even under YPD culture conditions. Under capsule-inducing conditions, a significant accumulation of vesicular structures (electron-lucent structures) was observed intracellularly, particularly in *ugg1Δ*. We performed lipid droplet (LD) staining to determine whether these vesicular structures might be LDs. We observed a notable increase in LDs in *ugg1Δ* under capsule-inducing culture conditions (*Figure 5—figure supplement 1A*). LDs impact proteostasis by sequestering misfolded proteins intended for degradation and providing an 'escape hatch' when the ERQC is overloaded (*Ploegh, 2007*; *Vevea et al., 2015*). Additionally, FM4-64 dye staining showed increased number of vacuoles in the *ugg1Δ* cells compared with that in the WT (*Figure 5—figure supplement 1B*). Moreover, all the observed abnormal phenotypes were less pronounced in the *mns1Δ101Δ* mutant than in the *ugg1Δ* strain. Collectively, the altered vesicular structures observed in the *ugg1Δ* and *mns1Δ101Δ* strains highlight the critical role of ERQC in maintaining proper vesicular architecture and cell surface organization. These functions are tightly linked to extracellular trafficking of virulence factors and cell wall remodeling.

## *ugg1Δ* transcriptomic profiling shows induced ER and cell wall integrity stress responses

To elucidate the mechanisms underlying *UGG1* deletion-induced physiological changes in *C. neoformans*, we performed RNA sequencing (RNAseq)-based transcriptome analysis of the WT and *ugg1Δ* strains under standard growth conditions (YPD medium, 30 °C). Comparative transcriptome analysis showed statistically significant alterations exceeding twofold in the expression patterns of 146 genes, of which 85 were upregulated and 61 were downregulated compared with that of the WT strain (*Figure 6A and B*, *Supplementary file 2A and B*).

Notably, the *ugg1Δ* mutant showed upregulation of genes such as *SKN1* and *KRE6* (transmembrane glucosidases involved in the sphingolipid biosynthesis and β-glucan biosynthesis), *CAT2* (a putative peroxisomal catalase), *CHS7* (chitin synthase export chaperone), and CNAG_05458 (a putative endo-1,3 (4)-β-glucanase). In contrast, genes encoding *ERG3* and *ERG25* (associated with the ergosterol biosynthesis pathway) and *FHB1* (flavohemoglobin denitrosylase associated with counteracting nitric oxide stress) were significantly downregulated. Gene ontology (GO) analysis in *ugg1Δ* showed pronounced induction of genes implicated in various cellular processes, including the hydrolysis of *O*-glycosyl compounds for cell wall remodeling, DNA replication, proteolysis, ribosome biogenesis,

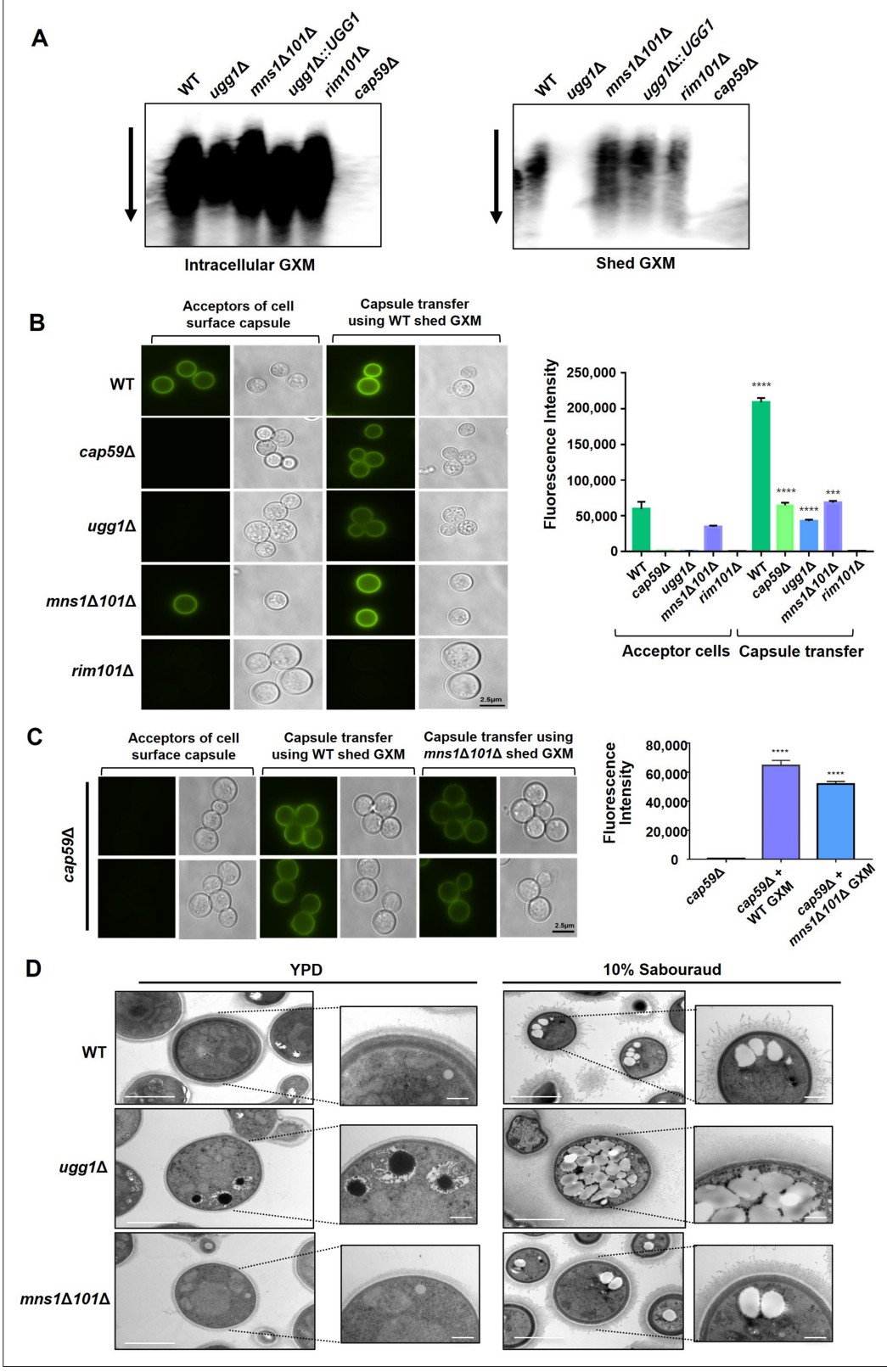

**Figure 5.** Capsule shedding and transfer analysis of *C. neoformans UGG1, MNS1*, and *MNS101* mutant strains. (**A**) Capsule shedding analysis. The presence of intracellular (left) and shed (right) glucuronoxylomannan (GXM) was assessed by blotting a cell culture filtrate using the monoclonal antibody 18B7. The arrow indicates the direction of electrophoresis. (**B**) Capsule transfer analysis using exogenous capsule material from WT. The capsule

*Figure 5 continued on next page*

*Figure 5 continued*

transfer assay was performed using the indicated strains as acceptors. Surface capsules were probed using the anti-GXM antibody 18B7 conjugated with AlexaFluor 488. Quantitative measurements of fluorescence intensity were calculated based on independent triplicate experiments with standard deviations presented as error bars. (**C**) Capsule transfer analysis using exogenous capsule material from WT or *mns1Δ101Δ*. Statistical significance: ****p<0.0001. All statistical data were determined based on one-way ANOVA and Dunnett's post hoc test. (**D**) Transmission electron microscopy (TEM) of *C. neoformans* WT, *ugg1Δ*, and *mns1Δ101Δ* strains. Yeast cells were grown overnight at 30 °C in YPD medium and fixed in 2% glutaraldehyde and 2% paraformaldehyde. A Zeiss Axioscope (A1) equipped with an AxioCan MRm digital camera was used to visualize India ink-stained *C. neoformans* cells. Specimens were prepared using critical point drying prior to TEM imaging. Capsule and yeast cell body diameters were measured using ImageJ (National Institute of Health).

The online version of this article includes the following source data and figure supplement(s) for figure 5:

**Source data 1.** Raw data related to *Figure 5*.

**Source data 2.** Uncropped membrane blot displayed in *Figure 5A* (left panel), indicating the relevant strains.

**Source data 3.** Uncropped membrane blot displayed in *Figure 5A* (left panel).

**Source data 4.** Uncropped membrane blot displayed in *Figure 5A* (right panel), indicating the relevant strains.

**Source data 5.** Uncropped membrane blot displayed in *Figure 5A* (right panel).

**Figure supplement 1.** Lipid droplets and vacuole staining of WT and *ugg1Δ* strains.

protein folding, and serine/threonine kinase activity. In contrast, genes associated with carbohydrate and lipid metabolism, chaperone binding, iron homeostasis, and mitochondrial intermembrane space were considerably downregulated (*Figure 6C*). The transcriptome profile strongly suggests that loss of Ugg1 function induces the expression of several genes involved in maintaining cell wall integrity to compensate for cell wall defects, particularly those associated with chitin (*CDA2*, *CHS7*, *QRI1*, and CNAG_06898) and glucan biosynthesis (*SKN1*, *KRE6*, *EBG1*, *LPI9*, CNAG_05458, and *BLG2*). Furthermore, genes coding for ER chaperones that aid in protein folding (*ERO1*, *KAR2*, *LHS1*, and *PDI1*) and chaperone regulator (*SCJ1*), and genes involved in proteolysis in the ER (CNAG_04635, CNAG_06658) were induced, likely as part of the ER stress response induced by the accumulation of misfolded proteins in the presence of a defective ERQC.

Further investigation of the effects of *UGG1* disruption in the expression of genes involved in capsule biosynthesis, cell wall remodeling, and the secretion pathways of protein virulence factors using qRT-PCR analysis (*Figure 6D*) showed no noticeable changes in the expression of genes related to capsule biosynthesis and conventional or unconventional secretion pathways at mRNA level, despite the aberrantly displayed defective phenotypes.

## EV-mediated protein trafficking and non-conventional secretion is impaired in the *ugg1Δ* mutant strain

Many enzymes contribute to the composite cryptococcal virulence phenotype. Some of these virulence-associated enzymes are secreted through traditional secretion pathways, whereas others are packaged into EVs and released into the extracellular milieu via non-conventional secretion mechanisms (*Almeida et al., 2015*). The defects observed in the ERQC mutant strains, particularly in the production of melanin and polysaccharide capsules, both of which are transported by EVs, suggest a possible impairment in the EV-mediated extracellular transport of virulence-associated enzymes in these mutants. Therefore, we examined the *in vitro* activity of urease, and laccase and acid phosphatase in both intracellular and extracellular fractions.

Comparative analysis of the WT and *ugg1Δ* strains showed minimal changes in the intracellular activities of urease, laccase, and acid phosphatase. However, their activities were significantly reduced in the extracellular fractions of the *ugg1Δ* cells (*Figure 7A–D*). Importantly, urease activity was performed in both solid and liquid media, demonstrating a notable loss of secreted urease activity in *ugg1Δ* cells, with a slight decrease in the *mns1Δ101Δ* double mutant cells (*Figure 7A and B*). As urease lacks a signal peptide required for conventional protein secretion, reduced urease secretion strongly indicates a compromised EV-mediated non-conventional secretory pathway in the *ugg1Δ* mutant strain. The decrease in extracellular laccase activity (*Figure 7C*) corroborates our melanization assay results, in which *ugg1Δ* showed defective melanin production (*Figure 4A*).

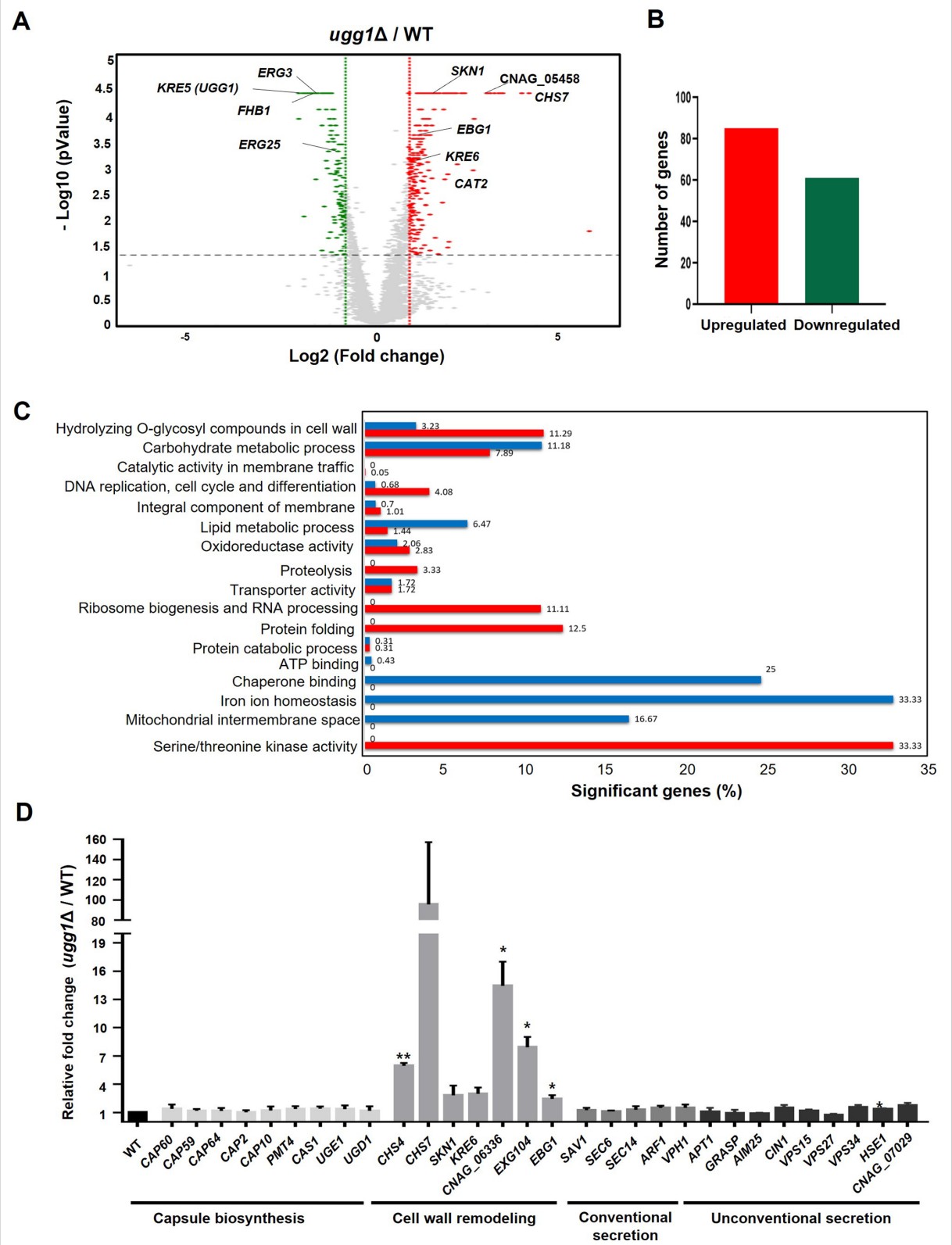

**Figure 6.** Transcriptome analysis of *C. neoformans* WT and *ugg1Δ* cells. (**A**) Volcano plot comparing a twofold differential gene expression between *ugg1Δ* and WT strains under normal growth conditions. (**B**) Number of genes upregulated and downregulated by ≥2-fold in *ugg1Δ* compared with that of the WT. (**C**) Gene ontology (GO) analysis of differentially expressed genes between WT and *ugg1Δ* strains. Significantly upregulated genes in *ugg1Δ* are shown in red, whereas significantly downregulated genes in *ugg1Δ* are shown in blue. (**D**) qRT-PCR analysis of mRNA expression levels of a set of

*Figure 6 continued on next page*

*Figure 6 continued*

genes responsible for capsule biosynthesis, cell wall remodeling, and both conventional and non-conventional secretion in *ugg1Δ* vs WT under normal growth conditions. Error bars represent standard deviation from three biologically independent experiment sets.

The online version of this article includes the following source data for figure 6:

**Source data 1.** Raw data related to *Figure 6*.

We next examined cellulase and α-amylase activities to evaluate the impact of ERQC disruption on the conventional secretion pathway of non-virulence-related enzymes (*Figure 7E and F*). The extracellular activities of these carbohydrate polymer-degrading enzymes in the *ugg1Δ* mutant were slightly lower than those observed in the WT, likely reflecting reduced intracellular enzyme production rather than a substantial impairment of secretion efficiency. Additionally, we analyzed the localization of chitin deacetylase I (Cda1), a glycosylphosphatidylinositol (GPI)-anchored protein involved in converting chitin to chitosan, essential for maintaining cell wall integrity (*Baker et al., 2007*; *Baker et al., 2011*; *Upadhya et al., 2018*; *Upadhya et al., 2021*). Cda1 was primarily detected in the insoluble cellular protein fraction, encompassing the cell wall and membrane compartments (*Figure 7G*, left), consistent with its cell surface localization via a GPI anchor. Cda1 is secreted extracellularly upon cleavage of the GPI anchor. Notably, the *ugg1Δ* mutant exhibited significantly higher intracellular and extracellular Cda1 levels compared to the WT and *mns1Δ101Δ* strains (*Figure 7G*, right), aligning with a 1.87-fold increase in *CDA1* mRNA expression observed in the RNA-seq data. These findings indicate that the surface localization and secretion of Cda1 remain efficient, suggesting that the conventional secretion pathway is largely unperturbed in *ugg1Δ*. Collectively, these results suggest that while Ugg1-mediated ERQC defects have a pronounced negative effect on EV-mediated protein transport, their impact on the conventional secretion pathway is minimal.

## Extracellular vesicle biogenesis and cargo loading are defective in *ugg1Δ* strain

We thus analyzed the number and size distribution of EVs to determine possible abnormal EV-mediated trafficking of virulence factors in the *ugg1Δ* mutant. Nanoparticle tracking analysis (NTA) showed a major peak in size distribution at approximately 150 (134 ± 28) nm in the WT strain, which was similar in size to mammalian exosomes and typical microbial EVs. Additionally, a minor peak ranging from 300 to 500 nm was observed and corresponded to microvesicles (*Figure 8A*). Notably, the size distribution of *ugg1Δ* EVs was more heterogeneous with smaller EVs ranging from 50 to 150 nm (82 ± 16; 124 ± 14) compared with that of the WT. The *cap59Δ* EVs displayed a major distribution of approximately 150 nm (120 ± 21). Microvesicles were barely detected in either *cap59Δ* or *ugg1Δ* strains.

We observed a significant reduction in the total number of secreted EVs in the *ugg1Δ* mutant (approximately 40%), suggesting defective EV biogenesis and/or stability in the absence of functional Ugg1 (*Figure 8B*). Cryo-TEM analysis of EV morphology confirmed that the sizes of EVs released by *ugg1Δ* were significantly smaller and more diverse than those of the WT or the acapsular mutant (*Figure 8C*), further supporting the NTA results. Examining EV size distribution by measuring their diameter (*Figure 8D*) further confirmed that although the EVs in the WT strain ranged from approximately 50–550 nm in size with distribution primarily concentrated at approximately 150 nm, the EVs in the *ugg1Δ* mutant were smaller (<100 nm) with heterogeneous sizes. Notably, *cap59Δ* EVs were uniformly distributed at approximately 150 nm and were present at a higher concentration than those of EVs from the WT. An increase in EV release and virulence factors has been also observed in acapsular mutant strains of *C. neoformans* and *C. gattii* (*Rodrigues et al., 2007*; *Reis et al., 2019*), supporting the notion that the capsule serves as a barrier in EV release in capsule forming species.

To characterize the protein composition of EVs from WT and *ugg1Δ* strains, EV-associated and whole-cell lysate (WCL) proteins were extracted and subjected to proteomic analysis (*Figure 8E*). In the WCL, 4,678 proteins were identified in the WT strain, with 333 exhibiting differential expression (>2-fold) in *ugg1Δ* vs the WT strain, including 123 upregulated and 210 downregulated proteins (*Figure 8—figure supplement 1A*, left). In EVs, 2,075 proteins were detected in the WT strain, of which 693 were differentially expressed in EVs from *ugg1Δ* relative to EVs from the WT strain (>2-fold), comprising 273 upregulated and 420 downregulated proteins (*Figure 8—figure supplement 1A*, right; *Figure 8—source data 1*). Additionally, comparative proteomic analysis revealed 180 proteins

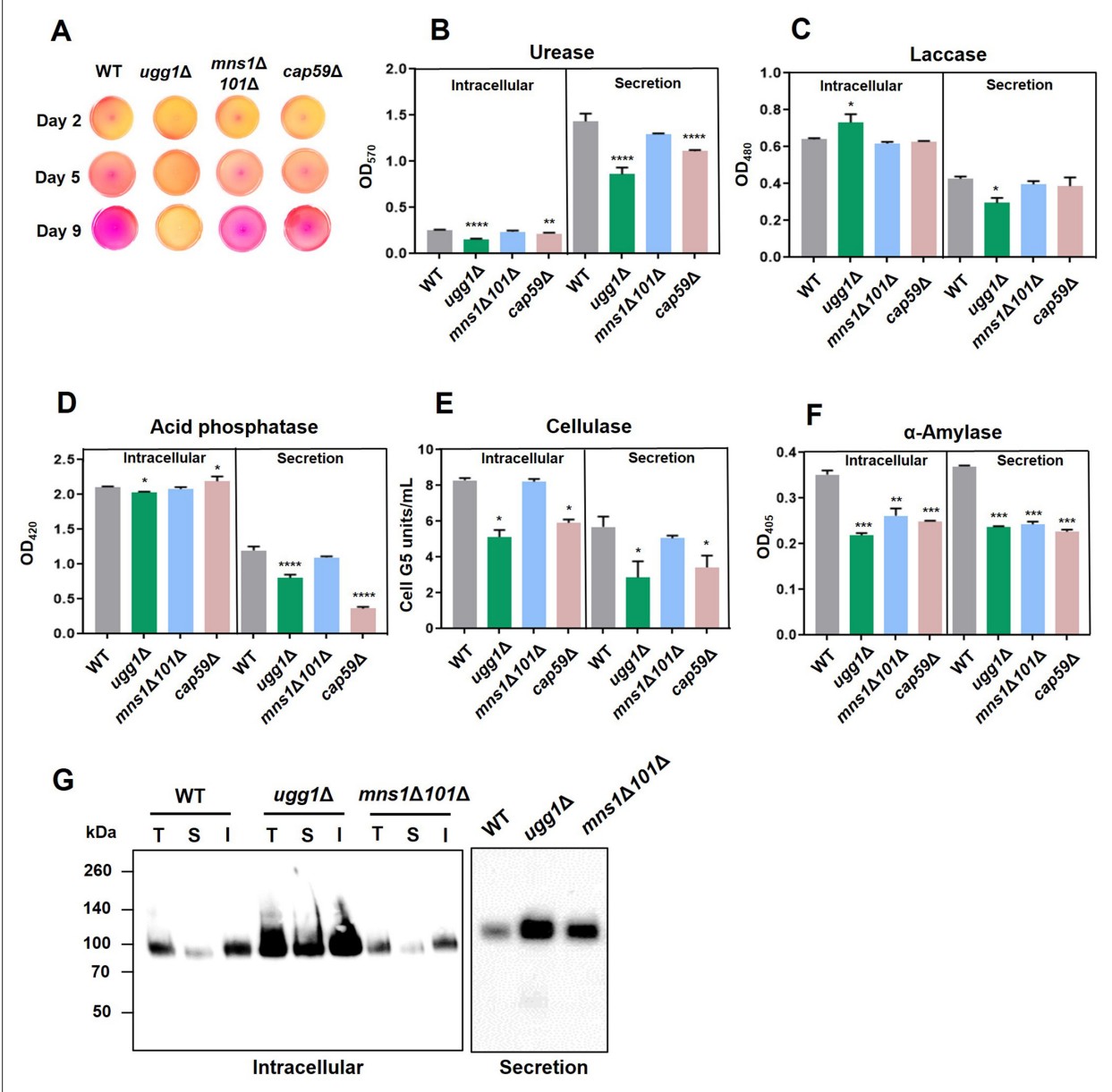

**Figure 7.** Analysis of protein secretion in *C. neoformans UGG1, MNS1*, and *MNS101* mutant strains. (**A**) Spot assay for urease analysis on Christensen's urea agar. Absence of pink coloration indicates loss of urease activity. (**B–D**) Analysis of secretion of the virulence-related enzymes urease, laccase, and acid phosphatase from three biologically independent experiment sets. Statistical significance: ****$p<0.0001$, **$p<0.003$, ***$p<0.005$, *$p<0.05$. All statistical data were determined based on one-way ANOVA and Dunnett's post hoc test. (**E, F**) Analysis of secretion for non-virulence-related enzymes such as cellulase and α-amylase. Statistical significance: ***$p<0.0005$, **$p<0.003$, *$p<0.05$, ns, not significant. All statistical data were determined based on one-way ANOVA and Dunnett's post hoc test. (**G**) Analysis of the conventional secretion of Cda1 in *C. neoformans*. Presence of Cda1 was analyzed in total (T), soluble (S), and insoluble (I) fractions of intracellular extracts (left), along with the secreted fraction (right). Subcellular fractionations were performed as previously described (*Thak et al., 2022*), and the fractions were subjected to western blotting analysis using an anti-Cda1 antibody.

The online version of this article includes the following source data for figure 7:

**Source data 1.** Raw data related to *Figure 7*.

**Source data 2.** Uncropped membrane blot displayed in *Figure 7G* (left panel), indicating the relevant strains.

**Source data 3.** Uncropped membrane blot displayed in *Figure 7G* (left panel).

**Source data 4.** Uncropped membrane blot displayed in *Figure 7G* (right panel), indicating the relevant strains.

**Source data 5.** Uncropped membrane blot displayed in *Figure 7G* (right panel).

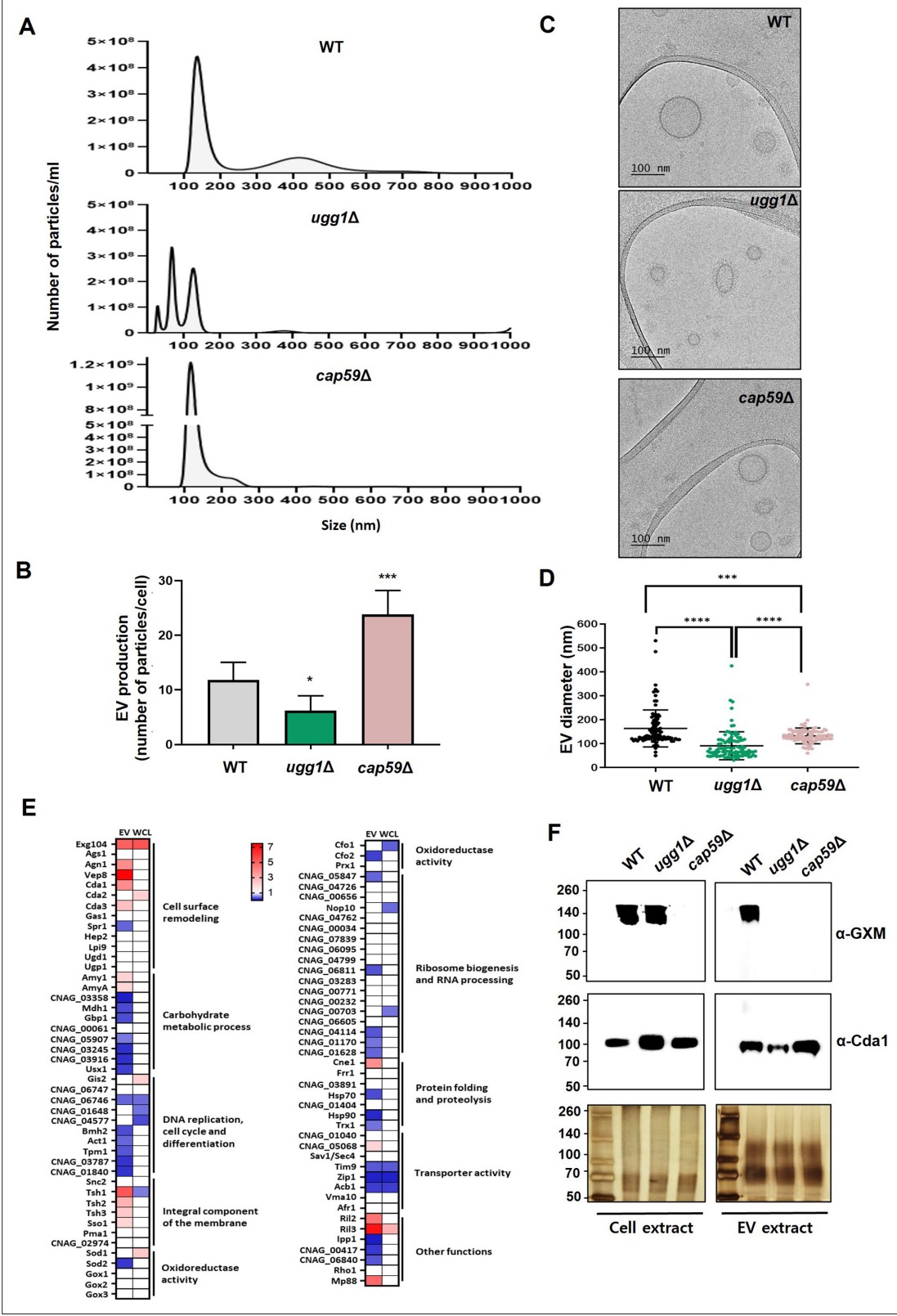

**Figure 8.** Analysis of extracellular vesicles (EVs) purified from WT, *ugg1Δ*, and *cap59Δ* cells. (**A, B**) Nanoparticle tracking analysis (NTA) of EVs extracted from WT, *ugg1Δ*, and *cap59Δ* strains and quantification of total EV concentration per cell density. Quantitative measurements were derived from three independent experiments with standard deviations presented as error bars. Statistical significance: ***p<0.0005, *p<0.05. All statistical data were determined based on one-way ANOVA and Dunnett's post hoc test. (**C, D**) Cryo-TEM imaging of purified EVs and comparative analysis of EV size in WT,

*Figure 8 continued on next page*

*Figure 8 continued*

*ugg1Δ*, and *cap59Δ* strains. Scale bar, 100 nm. The outer EV diameter of a total number of 100 EVs per strain, captured using cryo-TEM, were measured. (**E**) Heatmap representation of fold change between WT and *ugg1Δ* EV-associated proteins, commonly detected in this study and in previously reported EV proteome datasets (*ugg1Δ*/WT). Upregulated proteins in *ugg1Δ* are shown in red, whereas downregulated proteins are shown in blue. The proteome data of whole-cell lysates (WCL), generated from the cell pellets obtained after EV separation, were included for comparison. (**F**) Blotting analysis of GXM in the *C. neoformans* cells and EVs of WT, *ugg1Δ*, and *cap59Δ* strains. The 8 M urea extracts were obtained from EVs and cell pellets, from which EVs are generated. The urea extracts (5 µg total proteins) were loaded on 8% SDS-polyacrylamide gel and subjected to silver staining or blotting analysis using the anti-GXM 18B7 (α-GXM) and anti-Cda1 (α-Cda1) antibodies, respectively. Left: total cell extract. Right: total EV extract.

The online version of this article includes the following source data and figure supplement(s) for figure 8:

**Source data 1.** List of proteins identified in the proteomic analysis of the extracellular vesicles (EVs) and whole-cell lysates (WCL) obtained from wild type (WT) and *ugg1Δ*.

**Source data 2.** List of proteins identified in the proteomic analysis of the wild type (WT) and *ugg1Δ* secretome and whole-cell lysates (WCL).

**Source data 3.** Original gel displayed in *Figure 8F* (left panel), indicating the relevant strains.

**Source data 4.** Uncropped gels displayed in *Figure 8F* (left panel).

**Source data 5.** Original gel displayed in *Figure 8F* (right panel), indicating the relevant strains.

**Source data 6.** Uncropped gels displayed in *Figure 8F* (right panel).

**Source data 7.** Original gel displayed in *Figure 8F* (left, upper panel), indicating the relevant strains.

**Source data 8.** Uncropped gels displayed in *Figure 8F* (left, upper panel).

**Source data 9.** Original gel displayed in *Figure 8F* (left, middle panel), indicating the relevant strains.

**Source data 10.** Uncropped gels displayed in *Figure 8F* (left, middle panel).

**Source data 11.** Original gel displayed in *Figure 8F* (right, upper panel), indicating the relevant strains.

**Source data 12.** Uncropped gels displayed in *Figure 8F* (right, upper panel).

**Source data 13.** Original gel displayed in *Figure 8F* (right, middle panel), indicating the relevant strains.

**Source data 14.** Uncropped gels displayed in *Figure 8F* (right, middle panel).

**Source data 15.** Raw data related to *Figure 8*.

**Figure supplement 1.** Comparative proteomic analysis of whole-cell lysate (WCL) and extracellular vesicles (EVs) from WT and *ugg1Δ* strains.

**Figure supplement 2.** Comparative proteomic analysis of whole-cell lysate (WCL) and secretion fractions from WT and *ugg1Δ* strains.

uniquely enriched in EVs (*Figure 8—figure supplement 1B*). The GO enrichment analysis of differentially expressed proteins (>2-fold) showed distinctive patterns in biological process (BP) and cellular components (CC) between EVs and the WCL (*Figure 8—figure supplement 1C*). Three proteomic analyses of *C. neoformans* EVs have been reported previously (*Rodrigues et al., 2008*; *Wolf et al., 2014*; *Rizzo et al., 2021*), identifying 76, 202, and 1,847 proteins associated with EVs, respectively. We compared the EV protein content between WT and *ugg1Δ* identified in our study, focusing on overlapped proteins from the three EV proteomic data sets reported. Among 88 commonly reported EV-associated proteins, 29 were downregulated while 16 were overexpressed in *ugg1Δ* EVs compared to WT EVs (*Figure 8E*; *Supplementary file 3*). The differential expression pattern of the EV-associated proteins is quite distinctive from that of WCL, revealing significantly detectable changes in EVs. Thus, the absence of functional Ugg1 not only caused morphological alterations and reduced EV numbers but also altered EV protein abundance in *C. neoformans*.

To further examine possible cargo loading defects, we investigated the presence of GXM in the *ugg1Δ* EVs, comparing GXM quantities in the total cell extracts from WT, *ugg1Δ*, and *cap59Δ* strains (*Figure 8F*). Cda1 was used as the representative EV-associated protein as its presence was reported within the *C. neoformans* EV membrane, despite a substantial portion of Cda1 being secreted through the conventional secretion pathway (*Rizzo et al., 2021*). GXM polysaccharides were not detected in the EVs released from *ugg1Δ*, although it was present in the whole-cell extract. Taken together with the reduced quantity of EV-associated proteins, the absence of GXM polysaccharides in *ugg1Δ* EVs strongly indicates that EV cargo loading is defective when the ERQC is dysfunctional in *C. neoformans*.

## Discussion

As glycoproteins pass through the ER-Golgi secretory pathway, their *N*-glycans undergo extensive modifications in association with protein quality control. The ERQC system is highly conserved among eukaryotes but also diverged with distinct and species-specific features. The ERQC composition of *C. neoformans* is unique in that it possesses UGGT but lacks the three glucosyltransferases necessary for the addition of glucose residues to the core precursor *N*-glycans (*Park et al., 2012*). Additionally, it lacks CRT and carries multiple α1,2-mannosidases. In this study, we investigated the molecular features and functions of the *N*-glycan-dependent ERQC in *C. neoformans*, which generates unique *N*-glycan precursors without glucose addition, and shorter in length than those of most eukaryotes. Our data strongly suggest that despite the incomplete composition of the ERQC components, the UGGT-centered ERQC plays pivotal roles in cellular fitness, and particularly in the EV-mediated extracellular transport, which is crucial for pathogenicity.

The virulence of the *C. neoformans ugg1Δ* mutant was almost abolished in mice (*Figure 4D*), a phenotype consistent with the observed defects in key virulence determinants, such as the capsule and melanin, as well as poor growth at 37 °C (*Figure 3B*; *Figure 4A and B*). The *mns1Δ101Δ* mutant displayed intermediate phenotypes between those of the WT and *ugg1Δ*. Our data from the UPR induction analysis and comparative transcriptomic analysis strongly indicate that the ERQC mutation generates ER stress; this accounts for significant similarity of the defective phenotypes of the *C. neoformans* ERQC mutants to those reported in other mutants that exhibit defective protein folding in the ER, particularly in terms of increased stress sensitivity and decreased virulence. Connections between ER stress and thermotolerance have previously been established in *C. neoformans* as growth at 37 °C requires key ER protein chaperones and protein processing machinery, including components of the UPR signaling pathway Ire1 and the ER stress-responsive transcription factor Hxl1 (*Cheon et al., 2011*; *Havel et al., 2011*; *Jung et al., 2013*). Other mutations causing defects in ER function in *C. neoformans*, such as mutants lacking *DNJ1* (an ER J-domain containing co-chaperone) and *CNE1* (an ER chaperone), resulted in growth inhibition. The *dnj1Δ* mutant strain displayed impaired elaboration of virulence factors, such as the exopolysaccharide capsule and extracellular urease activity, when cultured at human body temperature (*Horianopoulos et al., 2021*). Altogether with our data from the ERQC mutants, these findings strongly support the notion that maintaining ER homeostasis is crucial for survival and virulence factor production at elevated temperatures.

It is quite notable that the *C. neoformans ugg1Δ* EVs exhibited defective loading of GXM and many protein cargo, alongside a decrease in the number and changes in the size distribution (*Figure 8*). Combined with compromised cell fitness, these defects in EV biogenesis and cargo loading, even under normal growth conditions, likely contribute to the complete loss of virulence owing to the defective export of virulence factors in the *ugg1Δ* mutant. Interestingly, our comparative proteome analysis of culture supernatants (secretome) revealed that the *ugg1Δ* strain exhibited more pronounced defects in the secretion of proteins lacking signal peptides, which are often linked to unconventional secretion mechanisms, compared to those with canonical secretion signals (*Figure 8—figure supplement 2*, *Figure 8—source data 2*). These findings further support the role of ERQC in modulating non-conventional secretion pathways.

It is intriguing how the absence of Ugg1, leading to ERQC defects, results in defective EV biogenesis and cargo loading in *C. neoformans*. In fungi, EV production and release occur through the maturation of endosomes into multivesicular bodies (MVBs), which are directed to the cell surface. Upon fusion with the plasma membrane, MVBs release vesicles into the extracellular environment, functioning as an unconventional secretion pathway (*Oliveira et al., 2010*). MVBs' formation relies on the functionality of the endosomal sorting complex required for transport (ESCRT), a highly intricate pathway involving a series of finely regulated events (*Henne et al., 2011*). Deletion of several ESCRT complex-related genes involved in EV transport directly impacts the cryptococcal capsule, notably resulting in reduced capsule size. Specifically, the *C. neoformans vps27Δ* strain showed altered EV size distribution, reduced capsule dimensions, defects in laccase export to the cell wall, and poor extracellular export of urease (*Park et al., 2020*). Other regulators of unconventional secretion are also linked to EV biogenesis. For example, the Golgi reassembly and stacking proteins (GRASPs) regulate EV cargo and dimensions in *C. neoformans* (*Peres da Silva et al., 2018*). A *graspΔ* mutant strain produced EVs with dimensions that significantly differed from those produced by WT cells, along with attenuated virulence and abnormal RNA composition. Moreover, the GRASP protein Grasp homology

1 (Grh1) serves as a chaperone that directly influences EV cargo (*Malhotra, 2013*; *Peres da Silva et al., 2018*). Autophagy regulators, which participate in EV formation in other eukaryotes, also play a role in cryptococcal EV formation. An *atg7Δ* strain manifests hypovirulence, and EVs produced by this strain show slightly different RNA composition compared with that of the WT cells (*Oliveira et al., 2016*). Notably, our data of the transcriptome and proteome analysis did not show significant changes in the ESCRT complex and GRASPs.

In addition to protein folding and secretion, the ER is crucial for regulating lipid metabolism (*Moncan et al., 2021*). Consequently, ER stress significantly impacts lipid and sterol synthesis, although some of these mechanisms are yet to be clarified. LDs aid the UPR and ERAD in degrading misfolded proteins during ER stress in *S. cerevisiae* (*Garcia et al., 2021*). In the present study, we observed a drastic increase in LDs in the *ugg1Δ* mutant (*Figure 5—figure supplement 1A*), consistent with the previous hypothesis that LDs help maintaining ER homeostasis. Furthermore, we observed increased number of vacuoles in *ugg1Δ* (*Figure 5—figure supplement 1B*,). This may thus indicate abnormal lipid homeostasis caused by the ERQC defects, which could, in turn, affect EV biogenesis. The importance of lipids and membrane regulators in proper EV formation and GXM export has been suggested in a previous study on the Apt1 flippase in *C. neoformans* (*Rizzo et al., 2018*). Additionally, ergosterol is essential for membrane fluidity, permeability, and protein transport (*Ermakova and Zuev, 2017*). Erg6 is involved in the ergosterol biosynthesis pathway and was identified as essential for the trans-Golgi network transport of proteins (*Proszynski et al., 2005*; *Nes et al., 2009*). In *C. neoformans*, the *erg6Δ* mutant released EVs with a significantly larger diameter than those of the WT, carrying increased levels of proteins and sterols, highlighting the role of ergosterol in cryptococcal EV biogenesis (*Oliveira et al., 2020*). Notably, the *ugg1Δ* mutant exhibited loss of microvesicles, which are derived from the plasma membrane (*Figure 8A*). Our preliminary analysis data on the surface lipid fraction of the *ugg1Δ* mutant indicated detectable alteration of sphingolipids and sterol profiles. Altogether, these findings suggest that ER stress caused by misfolded glycoprotein accumulation in ERQC-defective mutants may alter lipid composition, which could affect EV biogenesis.

A recent study using kidney cells presented strong evidence for a key role of ER stress in modulating EV biogenesis, demonstrating that ER stress decreases exosome production in mice (*Fukuoka et al., 2023*). They showed that T-cadherin is downregulated by ER stress through IRE1α activation at mRNA and protein levels and that ER stress decreases EV production through adiponectin/T--cadherin-independent way, which may involve interferon pathway activation in mice. We similarly observed induced activity of the ER stress sensor Ire1 in *ugg1Δ*, even under normal growth conditions, suggesting a possible association between the Ire1-mediated UPR pathway and EV biogenesis (*Figure 9*). Additionally, glycosylation regulates the biogenesis of small EVs and affects protein cargo loading efficiency in melanoma cells (*Harada et al., 2020*). This suggests that the altered *N*-glycosylation observed in *C. neoformans ugg1Δ* may influence cargo loading of certain EV-targeting glycoproteins. Despite their findings not being conducted in fungi, they provide valuable guides on the investigation of conserved mechanisms shared with *C. neoformans*. Further studies to investigate the mechanisms underlying the ERQC-mediated modulation of EV biogenesis and cargo loading in *C. neoformans* will provide insights into understanding the regulation, production, composition, and diversity of fungal EVs, enabling a better understanding of their biological function. Expanding our knowledge on pathogenic fungal EVs would pave the way for utilizing native or engineered EVs as promising candidates for therapeutic applications, including fungal infection diagnosis and vaccine development.

## Materials and methods
### Strains, culture conditions, plasmids, and primers
The *C. neoformans* strains and plasmids that were constructed and used in this study are listed in *Supplementary file 1A, B*). The plasmids and primers used are listed in *Supplementary file 1B and C*, respectively. The strains were typically cultured in YPD medium (1% yeast extract, 2% bacto peptone, and 2% glucose) at 30 °C with shaking (220 rpm). *C. neoformans* transformants were selected by culturing on YPD solid medium containing 100 μg/ml nourseothricin (Jena Bioscience, Germany; indicated as YPD$_{NAT}$), YPD solid medium with 100 μg/ml hygromycin B (Sigma-Aldrich, USA; indicated as YPD$_{HyB}$), or YPD solid medium with 100 μg/ml G418 (Duchefa, Netherlands; indicated as YPD$_{NEO}$). For

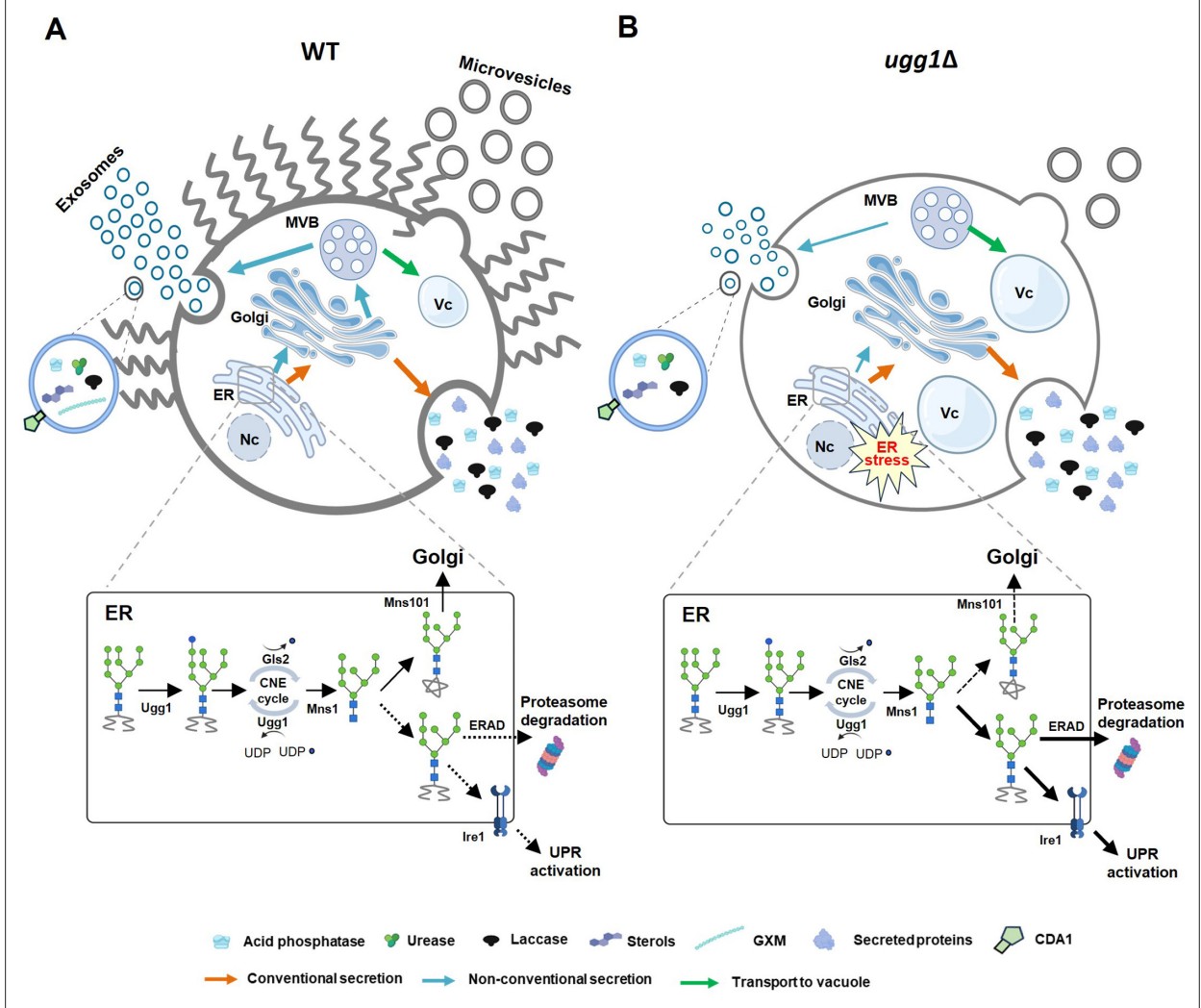

**Figure 9.** Impact of endoplasmic reticulum quality control (ERQC) disruption on glycoprotein folding and extracellular vesicle (EV)-mediated transport of virulence factors in *C. neoformans*. (**A**) In WT strain, *C. neoformans* UGGT homolog, Ugg1, functions as a sensor for misfolded glycoproteins within the ER, playing a crucial role in protein quality control. Functional ERQC is essential not only for ensuring the proper folding of glycoproteins, which is critical for maintaining cellular fitness, but also for facilitating EV-mediated secretion of capsule polysaccharides and virulence-related enzymes necessary for pathogenicity. (**B**) In the UGGT-deficient strain (*ugg1Δ*), ER stress is increased because of misfolded protein accumulation within the ER lumen. This heightened stress leads to decreased cellular fitness, which negatively impacts EV biogenesis and cargo loading. Consequently, significant defects occur in EV-mediated transport, which ultimately leads to a complete loss of virulence. Nc: nucleus; Vc: vacuoles. This figure was partially created using BioRender (https://BioRender.com/3gzzput, https://BioRender.com/bgibgte).

capsule induction, *C. neoformans* cells were cultured in liquid Sabouraud dextrose medium (Difco) at 30 °C for 16 h and incubated in 10% Sabouraud dextrose medium (pH 7.3) supplemented with 50 mM morpholinepropanesulfonic acid (MOPS) at 30 °C for 2 days. For melanin production analysis, the cells were spotted onto an L-DOPA agar plate (7.6 mM L-asparagine monohydrate, 5.6 mM glucose, 22 mM $KH_2PO_4$, 1 mM $MgSO_4 \cdot 7H_2O$, 0.5 mM L-DOPA, 0.3 mM thiamine-HCl, and 20 nM biotin) and incubated at 30 °C or 37 °C for 2 days.

## Construction of *C. neoformans ugg1Δ* and *UGG1* complementation strains

The *UGG1* deletion mutant strain was constructed by amplifying the DNA fragments containing the 5′- or 3′-flanking regions of CNAG_03648 ORF using PCR, in which genomic DNA from the H99 strain and the following primer sets 03648p_Fw/03648p_Rv and 03648t_Fw/03648t_Rv were used (*Supplementary file 1C*). The 5′- and 3′-regions of the selectable marker nourseothricin (NAT) were amplified

using the primer sets M13Fe/NSL-2 and M13Re/NSR-2 and pNAT-STM#159 as the template. *UGG1–NAT* fusion products of 5′- and 3′-flanking regions were generated using overlap PCR with the primer sets M13Fe/03648p_Rv and 03648t_F/NSR-2, respectively. The 5′- and 3′-fragments of the *UGG1* disruption cassette were introduced into the *C. neoformans* serotype A strain H99 through biolistic transformation. The transformants were selected on YPD$_{NAT}$, and gene disruption was screened using PCR. The complemented strain was generated by amplifying the DNA fragment containing *UGG1* using PCR, followed by subcloning into pJAFS1, containing the G418 resistance marker, via In-Fusion cloning (Takara Bio Inc, Japan). The resulting vector was excised at the unique EcoRV site and reintegrated into the native locus of the *ugg1Δ* strain through biolistic transformation. The transformants were selected on YPD$_{NEO}$, and gene disruption was screened using PCR.

## Construction of *C. neoformans mns1Δ*, *mns101Δ*, *mns1Δ101Δ*, and complementation strains

The *MNS1* and *MNS101* deletion mutant strains were constructed by amplifying the DNA fragments containing the 5′- or 3′-flanking regions of CNAG_02081 and CNAG_03240 ORFs by PCR using the genomic DNA from the H99 strain as a template and the primer sets: 02081_L1/02081_R1, 02081_L2/02081_R2 and 03240_L1/03240_R1, 03240_L2/03240_R2, respectively (*Supplementary file 1C*). The 5′- and 3′-regions of the selection marker HyB were amplified using the primer sets M13Fe/B5752 and M13Re/B5751 and using pJAF_HyG as the template. *MNS1-HyB* fusion products of 5′- and 3′-flanking regions were generated through overlap PCR using the primer sets M13Fe/02081_R1 and 02081_L2/NSR-2, respectively. *MNS101-NAT* fusion products of 5′- and 3′-flanking regions were generated through overlap PCR and the primer sets M13Fe/03240_R1 and 03240_L2/NSR-2, respectively. The 5′- and 3′-fragments of the *MNS1* and *MNS101* disruption cassettes were introduced into *C. neoformans* serotype A strain H99 through biolistic transformation. The transformants were selected on YPD$_{HyB}$ or YPD$_{NAT}$, respectively, and gene disruption was screened using PCR. The mutant strain lacking both *MNS1* and *MNS101* (*mns1Δ101Δ*) were generated by introducing the *MNS1* disruption cassette into the *mns101Δ* deletion strain. Each complemented strain was generated by amplifying the DNA fragment containing *MNS1* or *MNS101* using PCR, followed by subcloning into pJAFS1 via In-Fusion cloning (Takara Bio Inc). The resulting vector was excised at the unique BstBI site and reintegrated into the native locus of the *mns1Δ*, *mns101Δ*, or *mns1Δ101Δ* strains through biolistic transformation. The transformants were selected on YPD$_{NEO}$, and gene disruption was screened using PCR.

## Construction of *C. neoformans mnl1Δ*, *mnl2Δ*, and *mnl1Δmnl2Δ* and complementation strains

The *MNL1* and *MNL2* deletion mutant strains were constructed by amplifying the DNA fragments of the 5′- or 3′-flanking regions of CNAG_01987 and CNAG_04498 ORFs by PCR using genomic DNA from the H99 strain and the primer sets 01987_L1/01987_R1, 01987_L2/01987_R2 and 04498_L1/04498_R1, 04498_L2/04498_R2, respectively (*Supplementary file 1C*). The *MNL1* disruption was performed by amplifying the 5′- and 3′-regions of the selection marker NAT with the primer sets M13Fe/NSL-2 and M13Re/NSR-2 using pNAT-STM#159 as a template. The *MNL1-NAT* fusion products of the 5′- and 3′-flanking regions were generated using overlap PCR with the primer sets M13Fe/01987_L2 and 01987_L1/NSR-2, respectively. The *MNL2* disruption was performed by amplifying the 5′- and 3′-regions of the selection marker G418 with the primer sets M13Fe/B1887 and M13Re/B1886 using pJAFS1 as a template. The *MNL2-G418* fusion products of 5′- and 3′-flanking regions were generated using overlap PCR and the primer sets M13Fe/04498_R2 and 04498_F1/B1887, respectively. The 5′- and 3′-fragments of each *MNL1* and *MNL2* disruption cassettes were introduced into *C. neoformans* serotype A strain H99 through biolistic transformation. The transformants were selected on YPD$_{NAT}$ or YPD$_{NEO}$, respectively, and gene disruption was screened using PCR. Mutant strain lacking both *MNL1* and *MNL2* (*mnl1Δmnl2Δ*) were generated by introducing the *MNL2* disruption cassette into the *mnl1Δ* deletion strain. Each complemented strain was generated by amplifying the DNA fragment containing *MNL1* or *MNL2* using PCR and subcloned into pJAFS1. The transformants were selected on YPD$_{NAT}$, and gene disruption was screened using PCR.

## Construction and localization analysis of *C. neoformans* GFP-Ugg1, Mns1-GFP and Mns101-GFP fusion proteins

The GFP-UGG1 fusion vector expressing an N-terminal GFP fusion protein was generated by amplifying the *UGG1* ORF and the *C. neoformans* codon-optimized GFP sequences using the primer sets Bv_Not1_ugg1_pro_F1/SP_03648pro_R1, SP_CnGFP_F2/UGG1ORF_5gly_CnGFP_R2 and UGG1ORF_5gly_CnGFP_F3/Bv_Mfe1_UGG1ORF_R3, with pJAFS1_UGG1Com and pWH091 as templates. The fused PCR product was subcloned into the pJAFS1_UGG1Com plasmid via In-Fusion cloning (Takara Bio Inc, Japan) to generate the pJAFS1_SP_CnGFP_UGG1 plasmid. The resultant pJAFS1_SP_CnGFP_UGG1 plasmid was digested using EcoRV and biolistically transformed by integration into the CNAG_03648 locus of the *ugg1*Δ strain. The MNS1-GFP and MNS101-GFP fusion vectors expressing a C-terminal GFP fusion protein were generated by amplifying the *MNS1* and *MNS101* ORF sequences using the primer sets BamHI_MNS1 F1/MNS1_GFP R1 and BamHI_MNS101 F1/MNS101_GFP R1, respectively, and subcloned via In-Fusion cloning (Takara Bio Inc, Japan) into the pWH091 plasmid to generate the pWH091-MNS1 and pWH091-MNS101 plasmids. Both plasmids were digested with BstB1 and integrated into the CNAG_02081 and CNAG_03240 locus of the WT strain, respectively, by biolistic transformation.

The yeast cells were cultured in YPD medium for 24 h, and the obtained cell pellets were fixed in 4% paraformaldehyde (pH 7.0) for 10 min in rotation. The cells were washed twice with Phosphate-buffered saline (PBS) and stained with 1 µM ER Tracker red (Invitrogen, USA) for 20 min and 5 µg/ml of DAPI (4′,6-diamidino-2-phenylindole; Thermo Fisher Scientific, USA) for 10 min. The cells were washed twice with PBS and adjusted to an $OD_{600}$ of 1 in ultra-pure water. Fluorescence was observed through an Eclipse Ti-E fluorescence microscope (Nikon, Japan) equipped with a Nikon DS-Qi2 camera and a Plan Apo VC ×100 Oil DIC N2 (NA 1.4) lens. Images were processed on the NIS-elements imaging software (Nikon, Japan).

## HPLC and MALDI-TOF-based *N*-glycan structure analysis

cwMPs from *C. neoformans* were isolated and subjected to *N*-glycan structure analysis as described previously (*Park et al., 2012*; *Thak et al., 2018*). Briefly, *N*-glycans were released from purified cwMPs by PNGase F treatment (New England Biolabs, UK), followed by purification using a Carbograph Extract-Clean column (S*Pure, Singapore). The purified *N*-glycans were labeled with 2-aminobenzoic acid (2-AA; Sigma-Aldrich, USA) and further purified using a Cyano Base cartridge (Agilent, USA). 2-AA-labeled *N*-glycans were analyzed using a Waters 2690 HPLC system, equipped with a 2475 fluorescence detector, that was set to excitation and emission wavelengths of 360 nm and 425 nm, respectively. Data were collected using the Empower 2 software (Waters, USA). For MALDI-TOF analysis, neutral *N*-glycans were collected from HPLC fractionation and dried. The matrix solution was prepared as previously described (*Thak et al., 2018*) and mixed with the samples in equal volume. The samples were spotted on a MSP 96 polished-steek target (Bruker Daltonics, Germany), and the crystalized samples were analyzed using a Microflex mass spectrometer (Bruker Daltonics, Germany) in a linear negative mode.

## RNA preparation, qRT-PCR, and RNA-sequencing

Total RNA preparation, qRT-PCR, and RNA-sequencing were performed as previously described (*Thak et al., 2022*). Cells were inoculated in YPD medium at an $OD_{600}$ of 0.15 and cultured until it reached the mid-logarithmic phase ($OD_{600}$=0.5). Total RNA was extracted using the RNeasy Mini Kit (QIAGEN, Germany) and single stranded cDNA was synthesized using the Superior Script III reverse transcriptase (Enzynomics, South Korea). Reverse transcriptase-polymerase chain reaction (RT-PCR) was performed using serially diluted cDNA with the gene-specific primer sets (*Supplementary file 1, table 1C*) using Maxime PCR PreMix (i-Taq) (iNtRON Biotechnology, South Korea). Quantitative real-time PCR (qRT-PCR) was performed with the gene-specific primer sets (*Supplementary file 1, table 1C*) using a CFX96 Real-Time PCR detection system (Bio-Rad, USA). Normalized fold expression was calculated with the CFX manager software using the $2^{-\Delta\Delta CT}$ method, and with *ACT1* or *GAPDH* as the reference genes. For RNA-sequencing, the NEBNext Ultra II Directional RNA-Seq Kit (New England BioLabs, UK) was used to construct libraries. The mRNA was isolated using the Poly(A) RNA Selection Kit (Lexogen, Austria) and used for the cDNA synthesis. NovaSeq 6000 (Illumina) was used for high-throughput sequencing as paired-end 100 sequencing. Adapter and low-quality reads (<Q20) were

removed using FASTX_Trimmer and BBMap. Trimmed reads were mapped to the reference genome using TopHat, and the read count (RC) data were processed based on FPKM+ Geometric normalization method using EdgeR. Gene expression levels were quantified by evaluating the Fragments Per kb per Million (FPKM) reads values using Cufflinks. Data mining and graphic visualization were performed using ExDEGA (Ebiogen Inc, Korea).

## Animal study and *in vitro* survival analysis

Animal studies were conducted at the Chung-Ang University Animal Experiment Center. The study design was approved by the Ministry of Food and Drug Safety (South Korea). Survival and fungal burden were assayed as described previously (*Cheon et al., 2011*). Briefly, eight mice (6-week-old female A/J Slc mice; Japan) per strain were infected with $10^5$ cells via intranasal instillation. The mice were weighed and monitored once daily and euthanized after rapid 30% weight loss or identification of signs of morbidity. Kaplan–Meier survival curves were generated using Prism version 7 (GraphPad Software). For performing the fungal burden assay, the lungs of *C. neoformans*-infected mice were dissected on days 7 or 60. Half-organ portions of the excised lungs were homogenized, serially diluted, and plated onto YPD medium containing 100 µg/ml chloramphenicol (Sigma-Aldrich, USA). The other half-lung samples were fixed, sectioned, and stained with mucicarmine (Abcam, UK) for histopathological analysis. *C. neoformans* colonization was analyzed using a Zeiss Axioscope (A1) equipped with an AxioCam MRm digital camera.

Cell survival within macrophages (Korean Cell Line Bank, no. 40067) was analyzed by opsonizing *C. neoformans* cells with 10 mg/ml of 18B7 antibody at 37 °C for 1 h. Macrophages were seeded onto 96-well plates in Dulbecco's Modified Eagle Medium (DMEM) medium supplemented with 10% fetal bovine serum (FBS) and cultured at 37 °C in 5% $CO_2$ for 18 h. The opsonized *C. neoformans* ($10^5$) cells were co-incubated with activated macrophages at 37 °C in 5% $CO_2$ for 1 h. The non-phagocytized yeast cells were removed by washing each well thrice with PBS. Then, DMEM medium supplemented with 10% FBS was added to each well, followed by culturing at 37 °C in 5% $CO_2$ for 24 h. The macrophages were lysed in distilled water by vigorous pipetting, and the fungal cells were collected and serially diluted. Cryptococcal survival was assessed using two independent colony-forming unit (CFU) assays.

## Transmission electron microscopy

*C. neoformans* cells were cultured either at an initial $OD_{600}$ of 0.2 at 30 °C in YPD medium until $OD_{600}$ reached 0.8, or for 2 days in 10% Sabouraud media at 30 °C. The cell pellets were washed twice in PBS and fixed for 12 h in 2% glutaraldehyde-2% paraformaldehyde in 0.1 M phosphate buffer (pH 7.4). The cells were washed in 0.1 M phosphate buffer and post-fixed with 1% $OsO_4$ in 0.1 M phosphate buffer for 2 h. The cells were dehydrated using an ascending ethanol series (50, 60, 70, 80, 90, 95, 100%) for 10 min each and infiltrated with propylene oxide for 10 min. The specimens were embedded with a Poly/Bed 812 kit (Polysciences Inc, USA) and polymerized in an electron microscope oven (TD-700, DOSAKA, Japan) at 65 °C for 12 h. The block was cut into 200 nm semi-thin sections with a diamond knife in the ultramicrotome and stained with toluidine blue for observation using optical microscopy. The region of interest was further cut into 80 nm thin sections using the ultramicrotome, placed on copper grids, double stained with 3% uranyl acetate for 30 min and 3% lead citrate for 7 min, and imaged using a transmission electron microscope (JEM-1011, JEOL, Tokyo, Japan) equipped with a Megaview III CCD camera (Soft Imaging System, Germany) at the acceleration voltage of 80 kV.

## Capsule transfer and shedding analysis

Capsule transfer assays were performed as described previously (*Reese and Doering, 2003*). Briefly, conditioned medium (CM) was prepared as a source of GXM by culturing the respective strains for 5 days in YPD medium, followed by filtering and storing the culture supernatant at 4 °C. The acceptor strains were cultured overnight at 30 °C in YPD medium. In total, 2 × $10^6$ acceptor cells were incubated with 1 µl of CM for 1 h at room temperature under rotation (18 rpm) and washed twice with PBS. Capsule acquisition was visualized by incubating the cells with an anti-GXM (18B7) antibody conjugated with AlexaFluor 488 (Thermo Fisher Scientific, USA) for 1 h at 37 °C, and observing under an Eclipse Ti-E fluorescence microscope (Nikon, Japan), equipped with a Nikon DS-Qi2 camera. The images were processed using the NIS-elements microscope imaging software (Nikon, Japan).

Capsule shedding analysis was performed using a modified previously described protocol (*Yoneda and Doering, 2008*). The respective strains were cultured in 10% Sabouraud media for 2 days, and the culture supernatant was sterile filtered. Enzyme denaturation was performed by subjecting the filtrate to heating at 70 °C for 15 min, followed by centrifugation at 13,000 rpm for 3 min. Intracellular GXM analysis was performed by resuspending the pellets in TNE buffer (50 mM Tris-HCl [pH 7.5], 150 mM NaCl, 5 mM EDTA) with the same volume of glass beads (425–600 μm in diameter, Sigma-Aldrich, USA). The cells were disrupted four times for 15 s at 5000 rpm using a Precellys 24 Tissue Homogenizer (Bertin Technologies, France), followed by centrifugation at 16,000 rpm for 5 min at 4 °C. Next, 10 μl of the supernatant of either secreted or intracellular polysaccharide was mixed with 6× loading dye and run on a 0.6% certified megabase agarose (Bio-Rad, USA) gel in 0.5× TBE (44.5 mM Trisma base, 44.5 mM boric acid, 1 mM EDTA [pH 8.0]) at 25 V for 16 h. The polysaccharides were transferred onto a nylon membrane using the Southern blotting technique. The membrane was air-dried, blocked using 5% skim milk, and treated overnight with 2 μg/ml 18B7 antibody. After washing, the membrane was incubated with an anti-mouse peroxidase-conjugated secondary antibody (A0168, Sigma-Aldrich, USA) and subjected to detection using chemiluminescence.

## Detection of enzymatic activities in intracellular and secretary fractions

Biochemical enzymatic activities of acid phosphatase, urease, and laccase in the intracellular and secreted fractions were assayed spectrophotometrically using previously described methods (*Lev et al., 2014*; *Fu et al., 2018*; *de Sousa et al., 2022*). Acid phosphatase activity was assayed by culturing cells in MM-KCL medium (0.5% KCl, 15 mM glucose, 10 mM MgSO$_4$·7H$_2$O, 13 mM glycine, and 3 μM thiamine) for 3 h at 30 °C. The culture supernatant and soluble cell lysate were allowed to react with 2.5 mM p-nitrophenyl phosphate (pNPP) for 30 min at 37 °C. Urease activity was determined after culturing the cells in Rapid Urea Broth (RUH broth) for 24 h at 30 °C and subjecting the culture supernatant and soluble cell lysate to incubation with phenol red. Laccase activity was assayed by culturing the cells in asparagine salts media (7.6 mM L-asparagine, 0.1% glucose, 22 mM KH$_2$PO$_4$, 1 mM MgSO$_4$.7H$_2$O, 0.3 mM thiamine-HCl, and 20 nM biotin) for 48 h at 30 °C and allowing the supernatant and soluble cell lysate to react overnight with 10 mM L-DOPA. The reactions were quantified by measuring OD$_{420}$ (acid phosphatase), OD$_{570}$ (urease), or OD$_{480}$ (laccase). Cellulase activity was determined by culturing the cells in cellulose media (0.1% NaNO$_3$, 0.1% K$_2$HPO$_4$, 0.1% KCl, 0.5% MgSO$_4$, 0.5% yeast extract, 0.1% glucose, and 0.5% low-viscosity carboxymethyl cellulose) for 24 h at 30 °C. The culture supernatants were concentrated using an Amicon tube (30 kDa cutoff, Sigma-Aldrich, USA). The concentrated supernatants and soluble cell lysates were allowed to react at 40°C for 10 min with the substrate 4,6-O-(3-ketobutylidene)–4-nitrophenyl-β-D-cellopentaoside (BPNPG5), which was provided in the cellulase assay kit (CellG5 Method, Megazyme, Ireland). The reaction was terminated by adding 2% [w/v] Tris buffer (pH 10), and the absorbance of 4-nitrophenol was measured at 400 nm. Cellulase activity (CellG5 Units/ml) was calculated as indicated in the cellulase assay kit. α-amylase activity was measured using the α-Amylase Activity Colorimetric Assay Kit (Biovision Technologies, USA) according to the manufacturer's instructions. Yeast cells were cultured overnight in YPD medium at 30 °C, and the culture supernatants were concentrated using an Amicon tube (30 kDa cutoff, Sigma-Aldrich, USA). The concentrated supernatants and soluble cell lysates were incubated for 1 h at 25 °C in assay buffer with the substrate ethylidene-pNP-G7, which was provided in the kit. OD$_{405}$ was measured. All the activity analysis results were normalized according to cell density (OD$_{600}$).

## Subcellular fractionation and western blot analysis of Cda1

Subcellular fractionation was performed as previously described (*Thak et al., 2022*) for localization analysis of Cda1. Briefly, the respective strains were inoculated at an initial OD$_{600}$ of 0.5 in YPD medium and cultured at 30 °C for 16 h. Cell suspensions (OD$_{600}$=30) were divided into two tubes to extract total cellular proteins and to fractionate soluble/insoluble cellular proteins. The cells were disrupted four times using glass beads (425–600 μm in diameter; Sigma-Aldrich, USA) for 15 s at 5000 rpm in a Precellys 24 Tissue Homogenizer (Bertin Technologies, France). Total proteins were extracted by adding 5× sample loading buffer (62.5 mM Tris-HCl pH 6.8, 2.5% SDS, 0.002% Bromophenol Blue, 5% β-mercaptoethanol, and 10% glycerol) and boiling for 10 min. Cell debris and glass beads were removed by centrifugation for 1 min at 16,000 × *g*. The soluble protein fraction was obtained after centrifugation for 10 min at 16,000 × *g*, and 5× sample loading buffer was added to the supernatant,

followed by boiling for 10 min. The insoluble protein fraction was obtained by adding 1× sample loading buffer into the remaining pellets and boiling for 10 min. The obtained samples were adjusted to the same protein concentration and separated using SDS-PAGE. Cda1 expression was analyzed suing western blotting with an anti-Cda1 monoclonal antibody (*Upadhya et al., 2018*).

## Purification, nanoparticle tracking analysis (NTA), and cryo-TEM imaging of extracellular vesicles

EV purification was performed according to a previously published protocol (*Reis et al., 2019*) with modifications. One loop of cells was inoculated into 10 ml of liquid YPD and incubated at 30 °C for 24 h with shaking (220 rpm). The cells were washed twice with 10 ml of PBS, counted, and diluted in PBS to a density of $3.5 \times 10^7$ cells/ml. Aliquots of cell suspension (300 µl) were spread onto synthetic dextrose (SD) solid medium plates and incubated for 24 h at 30 °C. The cells were carefully recovered from each plate using an inoculation loop, gently resuspended in 30 ml PBS, and pelleted through centrifugation at 4000 rpm for 10 min at 4 °C. The supernatant was collected and centrifuged again at $15,000 \times g$ for 15 min at 4 °C. Then, the supernatant was filtered through 0.45 µm syringe filters and ultracentrifuged at $100,000 \times g$ for 1 h at 4 °C (MLA-50 fixed angle rotor, Beckman Coulter, Germany). The supernatant was discarded, and the EV pellets were collected and resuspended in 1 ml of PBS for immediate use or stored at –80 °C for further experiments. The samples were diluted 100-fold, and EV sizes were measured using an NTA instrument (NFEC-2024-03-295455, Nanosight Pro, Malvern Panalytical, Netherlands) coupled to a 532 nm laser (Malvern Panalytical, Netherlands), SCMOS camera (Hamamatsu Photonics. Japan), and syringe pump (Malvern Panalytical, Netherlands). The data were analyzed using the NS Xplorer software (v1.1.0.6, Malvern Panalytical, Netherlands).

Cryo-TEM imaging was performed by loading purified EVs (3 µl) onto a lacey carbon grid (Lacey Carbon, 300mesh Cu, Ted Pella Inc, USA), which was glow discharged at 15 mA for 60 s. The sample-loaded grid was blotted for 3 s at 15 °C and 100% humidity and immediately plunge-frozen in liquid ethane. The process was performed by Vitrobot Maek IV (Thermo Fisher Scientific, USA; SNU, CMCI). The frozen grids were imaged using TEM (JEM-2100F, JEOL); the temperature of the grid was maintained at approximately –180 °C at an acceleration voltage of 200 keV. The images were recorded using an ultrascan 1000 electron detector.

## Sample preparation for proteomic analysis

Proteins from EVs and WCL were prepared for proteomic analysis as follows. For EV proteome analysis, proteins were extracted by solubilizing purified EVs in a solution containing 8 M urea, 100 mM Tris (pH 7.5), and 5 mM Tris(2-carboxyethyl) phosphine (TCEP) for 20 min at 23 °C. For WCL samples, cell pellets collected after EV secretion were resuspended in the same extraction buffer and disrupted using glass beads (425–600 µm; Sigma-Aldrich, USA) in a Precellys 24 Tissue Homogenizer (Bertin Technologies, France) at 5000 rpm for four 15 s cycles. Following disruption, samples were centrifuged at 16,000 rpm for 5 min at 4 °C, and the supernatants were collected as WCL samples. For secretome and WCL proteome analysis, *C. neoformans* cells were cultured in SD liquid medium for 24 h. Culture supernatants were collected after centrifugation and filtered through 0.22 µm syringe filters to remove cellular debris, yielding the secretome samples. Cell pellets were resuspended in 8 M urea extraction buffer and disrupted using glass beads as described above to generate WCL samples.

Total protein concentration was quantified using the Protein Assay Dye (Bio-Rad, USA). Protein digestion was performed with Protifi S-Trap mini spin columns (C02-mini-80, Protifi, USA) and trypsin gold (V5280, Promega, USA). Digested peptides were eluted with 50 mM tetraethylammonium bromide (TEAB; Thermo Fisher Scientific, USA), 0.2% formic acid, and 50% acetonitrile. The pooled peptides were dried, dissolved in 100 mM TEAB, and labeled using the TMTpro 16plex Label Reagent Set (Thermo Fisher Scientific, USA).

## Proteomic analysis based on mass spectrometry and data processing

The labeled peptides (total, 100 µg proteins) were combined prior to offline basic reverse-phase liquid chromatographic (bRPLC) fractionation. Linear gradient was performed using buffer A (10 mM TEAB in water) and buffer B (10 mM TEAB in 90% acetonitrile), and 10 fractions were analyzed in total using an LC-MS/MS system from Korea Basic Science Institute (OC104). The samples were dissolved in 0.1% formic acid using an UltiMate 3000 RSLCnano system and analyzed using an Orbitrap Eclipse Tribrid

mass spectrometer (Thermo Fisher Scientific, USA). All MS raw files were converted into mzML and ms2 file formats using the MSConvert (version 3.0.20033) software. The *C. neoformans* proteome was determined by downloading a protein FASTA file from UniProt (http://uniprot.org), which included 7,492 reviewed (Swiss-Prot) and unreviewed (TrEMBL) proteins entries. A proteome search database with reversed sequence and contaminants in the Integrated Proteomics Pipeline (IP2, version 5.1.2, Integrated Proteomics Applications Inc, USA) was generated. The proteome search with 20 ms2 files was performed with IP2 and its following parameters, followed by evaluation for a false discovery rate using IP2 and Proteininferencer (version 1.0, Integrated Proteomics Applications Inc, USA). Protein quantification and statistical analysis for discovery of differentially expressed proteins were performed using the ms2 files with tandem mass tag reporter ions using an in-house program coded using Python 3.8, where *t*-test and Pearson's correlation analysis between comparison samples was performed using the scikit-learn (version 0.23.2), Scipy (version 1.6.0), and statsmodels (0.12.1) Python libraries.

## Acknowledgements

We thank Arturo Casadevall for providing the 18B7 antibody, Jennifer K Lodge for providing the anti-Cda1 antibody, and Ji-Yeon Kang for technical assistance with MALDI-TOF analysis.

## Additional information

### Funding

| Funder | Grant reference number | Author |
| --- | --- | --- |
| National Research Foundation of Korea | NRF-2022R1A2C1012699 | Hyun Ah Kang |
| National Research Foundation of Korea | NRF2018R1A5A1025077 | Hyun Ah Kang |
| National Research Foundation of Korea | RS-2023-00212663 | Eun Jung Thak |
| Korea Institute of Marine Science and Technology promotion | RS-2024-00405273 | Heeyoun Hwang |
| National Research Foundation of Korea | RS-2025-00521275 | Min-Ho Kang |
| National Research Foundation of Korea | RS-2025-00512716 | Hyun Ah Kang |

The funders had no role in study design, data collection and interpretation, or the decision to submit the work for publication.

### Author contributions

Catia Mota, Data curation, Validation, Investigation, Writing – original draft, Construction of mutants; Performing all virulence analysis; Kiseung Kim, Investigation, Construction of mutants; Performing phenotype analysis; Ye Ji Son, Validation, Investigation, Transcriptome analysis; Eun Jung Thak, Investigation, Glycan analysis; Su-Bin Lee, Investigation, Glycan analysis; Ju-El Kim, Investigation, NTA analysis; Jeong-Kee Yoon, Investigation, NTA analysis; Min-Ho Kang, Investigation, Cryo-TEM analysis; Heeyoun Hwang, Investigation, Proteome analysis; Yong-Sun Bahn, J Andrew Alspaugh, Data curation, Supervision, Writing – review and editing; Hyun Ah Kang, Conceptualization, Data curation, Supervision, Writing – original draft, Project administration, Writing – review and editing

### Author ORCIDs

Catia Mota ⓘ https://orcid.org/0009-0001-6129-0000
Kiseung Kim ⓘ https://orcid.org/0009-0004-8602-3293
Ye Ji Son ⓘ https://orcid.org/0009-0006-4324-5800
Eun Jung Thak ⓘ https://orcid.org/0000-0003-4523-142X
Ju-El Kim ⓘ https://orcid.org/0009-0002-3028-1125

Jeong-Kee Yoon https://orcid.org/0000-0003-0111-9911
Min-Ho Kang https://orcid.org/0000-0002-1342-0077
Heeyoun Hwang https://orcid.org/0000-0002-0124-2533
Yong-Sun Bahn https://orcid.org/0000-0001-9573-5752
J Andrew Alspaugh https://orcid.org/0000-0003-3009-627X
Hyun Ah Kang https://orcid.org/0000-0002-3722-525X

## Ethics

Animal studies were conducted at the Chung-Ang University Animal Experiment Center. The study design was approved by the Ministry of Food and Drug Safety (MFDS, South Korea).

Reviewer #2 (Public review): https://doi.org/10.7554/eLife.103729.3.sa1
Reviewer #3 (Public review): https://doi.org/10.7554/eLife.103729.3.sa2
Author response https://doi.org/10.7554/eLife.103729.3.sa3

---

# Additional files

## Supplementary files

Supplementary file 1. List of strains, plasmids and primers. (**A**) List of strains used in this study. (**B**) List of plasmids used in this study. (**C**) List of primers used in this study.

Supplementary file 2. List of *Cryptococcus neoformans* genes showing differential expression between *ugg1Δ* and wild type strains. (**A**) Upregulated (>2-fold) *Cryptococcus neoformans* genes in *ugg1Δ* than that of wild type (WT) cultivated in yeast extract peptone dextrose (YPD) at 30 °C. (**B**) Downregulated (>2-fold) *Cryptococcus neoformans* genes in *ugg1Δ* than that of wild type (WT) cultivated in YPD at 30 °C.

Supplementary file 3. Representative extracellular vesicle (EV)-associated proteins commonly detected in this study and previously reported *Cryptococcus neoformans* EV proteome data sets.

MDAR checklist

## Data availability

The raw RNA sequencing data have been submitted to the NCBI GEO database under accession no. GSE254772.

The following dataset was generated:

| Author(s) | Year | Dataset title | Dataset URL | Database and Identifier |
| --- | --- | --- | --- | --- |
| Mota C, Son YJ, Kim KS, Thak EJ, Lee SB, Kang HA | 2025 | Unraveling the Evolutionary Unique Glycoprotein Quality Control System and its Roles in Cellular Fitness and Extracellular Vesicle Transport in *Cryptococcus neoformans* | https://www.ncbi.nlm.nih.gov/geo/query/acc.cgi?acc=GSE254772 | NCBI Gene Expression Omnibus, GSE254772 |

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
