## [Editor Report · eLife Assessment]

This **important** study confirms the molecular function of putative components of the N-glycan-dependent endoplasmic reticulum protein quality control (ERQC) system in the pathogen *Cryptococcus neoformans*. The study demonstrates an involvement in fitness, virulence, and the secretion and composition of extracellular vesicles, albeit in ways that are not yet fully understood. The evidence provided is **convincing**, with rigorous, well-controlled assays and the use of complemented strains.

---

## [Referee Report · Reviewer #2 (Public review)]

Summary:

This study investigates the molecular function of the N-glycan-dependent endoplasmic reticulum protein quality control system (ERQC) in *Cryptococcus neoformans* and correlates this pathway with key features of C. neoformans virulence, especially those mediated by extracellular vesicle transport. The findings provide valuable insights into the connection between this pathway and the biogenesis of C. neoformans extracellular vesicles.

Strengths:

The strength of this study lies primarily in the careful selection of appropriate and current methodologies, which provide a solid foundation for the authors' results and conclusions across all presented data. All experiments are supported by well-designed and established controls in the study of C. neoformans, further strengthening the validity of the results and conclusions drawn from them. The study presents novel data on this important pathway in C. neoformans, establishing its connection with C. neoformans virulence. Interestingly, the findings led the authors to understand the relationship between this pathway and the transport of key fungal virulence factors via extracellular vesicles. This was demonstrated in the study, paving the way for a deeper understanding of extracellular vesicle biogenesis-a field still filled with gaps but one that this study contributes to with solid data, helping to clarify aspects of this process.

---

## [Referee Report · Reviewer #3 (Public review)]

Summary:

*Cryptococcus neoformans* is a global critical threat pathogen and the manuscript by Mota et al demonstrates that the pathogen's N-glycan-dependent protein quality control system regulates the capacity of the fungus to cause disease. The system ensures that glycoproteins are folded correctly. The system is involved in fitness and virulence of the fungus by regulating aspects of cellular robustness and the trafficking of virulence-associated compounds outside of the cell via transport in extracellular vesicles.

Strengths:

The investigators use multiple modalities to demonstrate that the system is involved in cryptococcal pathogenesis. The investigators generated mutant C. neoformans to explore the role of genes involved in the protein folding system. Basic microbiology, genetic analyses, proteomics, fluorescence and transmission microscopy, nanotracking analyses, and murine studies were performed. The validity of the findings are thus very high. Hypotheses are robustly demonstrated.

---

## [Author Response]

The following is the authors’ response to the original reviews

**Reviewer #1 (Recommendations for the authors):**
Major comments(1) The section on page 20 describing the proteomic analysis of EVs is poorly written and confusing, with a lot of data in the supplement. It is not clear what the proteomics data actually means.

We appreciate your feedback on the clarity of the proteomic analysis section. We have rewritten the section on page 20 with more detained information to provide a clearer explanation of the proteomics data and its biological significance. Additionally, we have incorporated a comparative analysis of the EV and total cell lysate proteomes (Fig. 8E, Supplementary Fig. S7A, Supplementary Tables 3 and 4) for supplemental data interpretation.

(2) The order of the data could be improved.

We appreciate your feedback regarding the data organization. We have reorganized the order and position of some data in a more structured and coherent manner, as suggested by the reviewers.

- Reorganization of the qPCR data (previously Fig. 1C) as Fig. 3A

- Removal of the data on the growth analysis on raffinose media (previously Fig. 7H).

-Reorganization of the spotting data of the double mutant (previously Fig 3B) to Supplementary Fig. S3B

- Reorganization of the subcellular localization data (previously Fig 3E) to Supplementary Fig. S4A

(3) The discussion is repetitive with the introduction and merely summarizes the results and speculates on the mechanism of how the absence of UGGT, leading to ERQC defects, results in defective EV biogenesis/cargo loading in C. neoformans.

We removed several repetitive sentences in the discussion and provided additional information on proteome analysis.

Other questions and comments(1) Instead of comprehensively analyzing EVs from the UGG1 mutant, a more informative approach to better understanding how defects in N-linked glycosylation impact secretion, would be to do a proteomic analysis on the total secretions (including beta glucanase-treated cells to release classically secreted proteins from the cell wall) and EVs.

We agree that a comprehensive proteomic analysis of total secretions and classically secreted proteins would provide deeper insights into how defects in *N*-glycosylation impact secretion in *C. neoformans*. To address this concern, we performed an additional set of proteomic analyses, the proteome profiles of total cell lysates and the secretome of *C. neoformans* cultivated in SD broth and presented the results as Supplementary Table S5 and Supplementary Fig. S7B. These additional analyses provide further insights into the impact of *UGG1* deletion on both conventional and unconventional secretion pathways, supporting a more pronounced effect of the *UGG1* defect on EV-mediated trafficking. The discussion has been updated accordingly (Page 22, lines 509-514).

(2) The melanization defect in Ugg1 mutant is not strong. Could the reduction be due to partially compromised Ugg1 mutant growth at 30{degree sign}C as indicated in the spot tests. Were photos of the spot dilution assays taken at 1 and 2 days to investigate slower growth? Or alternatively were growth curves taken in a liquid culture?

For accuracy of melanin synthesis defect, in addition to analysis on L-DOPA plates, we had assessed melanin production in liquid L-DOPA medium following a 3-day incubation, and the melanin production in liquid media was normalized by cell density (OD_600_). The data on normalized melanin production is now included as Fig. 4B in the revised manuscript. The defective laccase activity in the *ugg1*Δ mutant (Fig. 7C) further corroborates our melanization assay results, which is additionally mentioned in the text (Page 18, lines 393-395).

(3) Is it accurate to say that some virulence factors (i.e. melanin, capsule and phosphatases) are predominantly trafficked through EV's in C. neoformans? Have studies been done to determine the proportion of virulence factors trafficked via EV's versus traditional secretion?

We thank you for the thoughtful comments. Some virulence factors, such as urease, melanin and capsule polysaccharides, lack a signal peptide required for targeting for the conventional ER/Golgi secretion pathway. It is generally assumed that the trafficking of these factors in *C. neoformans* is predominantly mediated by non-conventional secretion via EVs. Additionally, even some virulence factors with signal peptides, such as laccase and phosphatases, are also transported via EVs besides the conventional secretion. The quantitative analysis to compare the proportion of virulence factors secretion via EVs versus the conventional pathway has not been yet reported, despite that genetic evidence suggests that conventional secretion also plays a significant role in the export of capsule polysaccharides. Thus, we were also careful not to highlight EV as the main route of virulence factors in the manuscript.

(4) There is insufficient background in the introduction linking what is known about the ERQC process to secretion in general. The topic changes from the ERQC process to fungal virulence factor, with a primary focus on non-classical (EV-based) secretion. Classical secretion should also be discussed without assuming that non classical (EV) secretion is the major pathway contributing to fungal virulence.

We appreciate your insightful comments highlighting the need for more background on the ERQC process and its relationship with secretion. To address the reviewer’s concerns, we have added sentences to describe the key roles of ERQC in conventional protein secretion in the Introduction (Page 5, lines 102-106).

(5) Figure 1A. What does the blue filled circle with the red outline signify? Fig 1 A legend is not well explained. A summary using material provided in the intro/discussion should be included to briefly explain the process and the differences between fungal species. Please also be aware that the intro starts describing the human ERQC process and then switches to what happens in *S. cerevisiae*.

We have revised Figure 1A by removing the red circle and updated the figure legend in the revised manuscript to include more detailed information about the ERQC differences across higher eukaryotes and fungal species.

(6) Figure 2A. There are no units on the Y-axis. Presumably, the scale is the same for all 3 strains.

Thank you for your comments. The Y-axis is the same for all three strains and, as in Fig. 2C, and represents the relative fluorescence intensity obtained from the HPLC analysis. We added the units on the Y-axis in Fig. 2A.

(7) If Mnl1 and 2 have proposed roles in proteasomal degradation, wouldn't they be expected to have ER retention signals, like Ugg1?

We appreciate your valuable insights regarding the absence of ER retention signals in Mnl1 and Mnl2. Previous studies have shown that *Saccharomyces cerevisiae* Mnl1/Htm1 does not possess canonical KDEL/HDEL-like ER retention signals. Instead, its retention in the ER lumen is facilitated through its interaction with protein disulfide isomerase Pdi1, which contains an HDEL sequence (Gauss et al. 2011). Thus, it is expected that non-canonical retention mechanisms—such as interactions with other ER proteins—could contribute to the retention of Mnl1 and Mnl2 within the ER. We added this information to the revised manuscript (Page 8, lines 154-159).

(8) Figure 1 C qPCR showing change in mRNA in response to ER stress should not be grouped in this figure. It could be standalone or discussed when the spot dilution assays are performed. Anyway, spots tests are more convincing of a role in stress response than qPCR as the ugg1 mutant is sensitive to tunicamycin, DTT and cell wall stressing agents.

As suggested by the reviewer, we have reorganized the qPCR data as a part of Figure 3 (Figure 3A) in the revised manuscript.

(9) It is odd that mns1/101 mutants are not sensitive to ER and CW stress given their proposed differing location/function in the pathway (Figure 1) determined from the N-linked profiling. Any explanation? Could there be redundancy?

We appreciate the reviewer’s observation regarding the lack of ER and CW stress sensitivity in the *mns1*Δ and *mns101*Δ mutants, despite their proposed roles in *N*-glycan processing. We had previously reported that the *C. neoformans alg3*Δ mutant, lacking a critical enzyme responsible for the synthesis of Dol-PP-Man_6_GlcNAc_2_ in the *N*-glycosylation pathway, exhibited clearly impaired *N*-glycan elongation, but showed no detectable growth defects even under stress conditions *in vitro*. However, *alg3*Δ is avirulent in *in vivo* pathogenicity (Thak et al., 2020). Similarly, the *mns1*Δ*101*Δ double mutant shows glycan-processing defects that do not compromise cellular fitness under stress conditions but result in attenuated virulence in animal models. These findings suggest that some glycosylation-related defects may impact more severely *in vivo* pathogenicity rather than *in vitro* stress sensitivity.

(10) Although the Silver-stained gels of the ugg1 mutant are not particularly informative, why weren't they (and Con A blots) performed for the other mutants?

The overall decrease of hypermannosylated glycans observed in the *ugg1*Δ mutant allowed us to detect clear alterations in protein glycosylation patterns in the lectin blot using *Galanthus nivalis* agglutinin, which recognizes terminal α1,2-, α1,3-, and α1,6-linked mannose residues. In contrast, the limited changes of a few glycan species in other mutants, including *mns1*Δ, *mns101*Δ, and *mns1*Δ*101*Δ, are relatively subtle to be detected in the lectin blot, due to only minor differences in the average lengths of their *N*-glycans compared to the WT. Therefore, we presented the lectin blotting data only for the *ugg1*Δ mutant.

(11) If there is ER stress under normal conditions in the Ugg1 mutant then technically this mutant should be growing more slowly under normal conditions. This is difficult to predict in a spot dilution assay where growth is only visualized at day three when any growth defect may have been corrected. The slower growth rather than the reduced secretion of GXM specifically is therefore more likely to be responsible for the reduced virulence.

We appreciate the reviewer’s insightful comment regarding the interplay between ER stress, growth defects, and virulence attenuation in the *ugg1*Δ mutant. While retarded growth in *C. neoformans* is often associated with reduced virulence, there are a few exceptions. For instance, disruptions in cell cycle progression in *C. neoformans* have been reported to result in larger capsule sizes, which rather enhance *in vivo* virulence when analyzed in *Galleria mellonella* infection models (García-Rodas et al., 2014). This highlights that growth defect alone is not sufficient for virulence attenuation. In the case of the *ugg1*Δ mutant, we speculate that the almost complete loss of virulence is attributed not only to its growth retardation but also to its impaired secretion of key virulence factors, including the polysaccharide capsule.

(12) The rationale for using leucine analogue 5',5',5'-trifluoroleucine (TFL), in a growth assay (Fig. 3C) to determine whether the defective ugg1Δ phenotypes are induced by ER stress caused by misfolded protein accumulation is not explained.

The leucine analogue 5',5',5'-trifluoroleucine (TFL) can be incorporated into newly synthesized proteins, disrupting normal folding and thus leading to the generation of misfolded proteins (Trotter et al., 2002; Cowie et al., 1959). In the context of a defective ERQC pathway, these misfolded proteins cannot be adequately repaired, resulting in their accumulation and triggering ER stress. Excessive ER stress may ultimately inhibit cell growth in the presence of TFL. This explanation has been incorporated into the revised manuscript (Page 11, lines 236–241).

(13) I would argue that only the Ugg1 and double Mns mutant were defective in virulence. For the single mutants, it looks like no difference was found relative to WT. The longer median survival of these mutants (if significant) is most likely due to poor infection technique.

We agree with the reviewer’s opinion that the *mns1*Δ and *mns101*Δ single mutants have no significant difference in *in vivo* virulence compared to the WT strain, unlike the *mns1*Δ*101*Δ double mutant which showed significant attenuated virulence. We had previously addressed that in the manuscript (Page 13, lines 267-269).

(14) The authors conclude that the ugg1Δ strain specifically is impaired in extracellular secretion of capsular polysaccharides but is this via classical (SAV1) secretion or EVs?

In addition to EV-mediated transport, capsular polysaccharide secretion can occur via the Sav1 (Sec4p)-mediated classical secretion pathway. However, our proteome data of total cell lysates indicated that the protein levels of Sav1 were comparable between the WT and *ugg1*Δ strains, suggesting that Sav1p function itself might not be impaired. Given that the *ugg1*Δ mutant exhibits altered vesicular structures (Supplementary Fig. S6) and loss of microvesicles (Fig. 8A), we speculate that a defect might occur at a post-Sav1p step, such as vesicle fusion with the plasma membrane, likely contributing to the complete defect in secretion of capsular polysaccharides in the *ugg1*Δ strain, in which EV biogenesis and defective cargo loading are severely impaired, producing EVs that lack capsular polysaccharides (Figure 8F). However, further studies should be carried out to define the contribution of SAV1 to the secretion of capsular polysaccharides in in the *ugg1*Δ strain.

(15) The rationale for doing 7 H is very confusing.

The experiment assessing raffinose utilization as a carbon source was inspired by the previous work of Garcia-Rivera et al., reporting that the *cap59*Δ mutant is unable to utilize raffinose due to a defect in the secretion of raffinose-hydrolyzing enzymes. As another way to investigate potential defects in the conventional secretion pathway, we investigated the growth of the *ugg1*Δ mutant in the presence of raffinose. Due to our extensive data length, we have decided to remove this complementary data from the manuscript.

(16) It is speculated in the discussion that ER stress impacts lipid/sterol synthesis and that LDs (lipid droplets?) aid the UPR and ERAD in degrading misfolded proteins during ER stress in *S. cerevisiae*. The authors mention that they observed a drastic increase in LDs in the ugg1Δ mutant. Where is this data? Even with the data, this is all speculation. The authors also speculate that increased numbers of vacuoles in ugg1 (where is the data?) could be the cause of the altered vesicular structures observed in the mutants, which may indicate abnormal lipid homeostasis caused by the ERQC defects, which could, in turn, affect EV biogenesis. Again, this is speculative.

The data on lipid droplets (LDs) and vacuole staining are presented in Supplementary Figure S6, showing a drastic increase in LDs and an increased in vacuolar size in the *ugg1*Δ mutant compared to the wild-type strain, especially in capsule-inducing conditions. In addition to such changes in vesicular structures, our preliminary data on sphingolipids and sterol analysis in the surface lipid fraction of the *ugg1*Δ mutant led us to propose the hypothesis that ERQC defects may impact lipid metabolism, which in turn could influence EV biogenesis and membrane properties. It is expected that these findings would provide a strong foundation for future studies exploring the link between ERQC, lipid homeostasis, and EV biogenesis. We have revised our speculation on the association of abnormal lipid homeostasis, caused by ERQC, with EV biogenesis more appropriately by adding the information on our preliminary data of lipid profiles and mentioning that the *ugg1*Δ mutant lacks microvesicles, which are derived from the plasma membrane (Page 24, lines 554-559).

**Reviewer #2 (Recommendations for the authors):**
(1) My suggestions for the authors are the same as those presented in the public review: (1) reducing the text in certain sections of the paper to improve readability for the audience, and (2) reconsidering the figures to reduce the amount of information in each one, moving some of the content to the supplementary material.

We thank the reviewer for their constructive suggestions regarding the organization and readability of the manuscript. As suggested, we addressed your concerns as follows:

(1) Reducing the text in the Introduction, Results, and Discussion sections by removing repetitive statements and simplifying complex descriptions where possible.

(2) Changing the presentation of figures: we have also reorganized the presentation of some data by moving non-essential data to the supplementary material. The updated figures and supplementary materials have been clearly referenced in the text to guide readers.

(3) Reorganization of materials and methods: some parts of methods were moved to Supplementary Information

(4) Removal of Figure 7H and the sentences describing the result

More detailed explanations on the reduction and reorganization are also described in the response to the major comments (2) and (3) made by Reviewer #1.

(2) Figure 3, for example, shows no difference in fungal growth under different cultivation conditions. This information is valuable but could be mentioned in the text, with the image provided as supplementary material, focusing the figure only on images that show significant growth differences among the strains. I suggest a similar approach for other figures so that the authors can include only the most relevant results in the main body of the article and move some figures to the supplementary materials.

For Fig. 3, the spotting data of the double mutant (previously Fig. 3B) is now presented in the supplementary information (Supplementary Fig. S3B). Additionally, the subcellular localization data (previously Fig 3E) was also moved to the supplementary material (Supplementary Fig. S4A).

**Reviewer #3 (Recommendations for the authors):**
(1) Line 43 "EV-mediated transport of virulence bags" doesn't make sense. EVs have been described as "virulence bags" (and are in this work later in the introduction) but this should here be "transport of virulence factors" or "compounds associated with virulence" but only if you have confirmed that the "cargo" is consistent with this- which is not evident in the abstract.

Thank you for your insightful comment. We have revised this to "EV-mediated transport of virulence factors" in line with your suggestion.

(2) Line 49 "secretory pathway" - is there not more than one secretion pathway?

Thank you for pointing this out. The term "secretory pathway" has been updated to "secretory pathways" to acknowledge the presence of both conventional and unconventional secretion mechanisms.

(3) Line 53 "recognizes folding defects, repairs them, and ensures the translocation of irreparable misfolded proteins" should be "recognizes folding defects and repairs them or ensures the translocation of irreparable misfolded proteins.

Thank you for pointing this out. We have revised the sentence as you suggested.

(4) Lines 88-90 ALG needs to be written out the first time - Asn-linked glycans. Also, consider adding that ALG genes are present in most eukaryotes as it is unclear what you are comparing C. neoformans to.

Thank you for your helpful comment. We have revised the text to write out "ALG" as "Asn-linked glycosylation" and added the sentence “*ALG* genes are evolutionary conserved in most eukaryotes” in the revised manuscript (Page 4, line 84).

(5) Line 99 Cryptococcus has already been abbreviated to C. so don't write it out again.

We have corrected "*Cryptococcus*" to “*C.*” throughout the manuscript after its first mention.

(6) Line 152- tunicamycin and DTT are not described yet, which may make it challenging for some readers to understand what these drugs are doing/why they were used. What is on lines 156 and 157 for these drugs should go up with the first mention of these drugs.

Thank you for your helpful suggestion. We have revised the manuscript to include the descriptions and purpose of using tunicamycin (TM) and dithiothreitol (DTT) immediately following their first mention, as recommended (Page 10, lines 208-210).

(7) The text for Figure 1 C is inaccurate. High temperature also induced KAR2, as noted above, but inaccurately stated in line 160. There is no comment on the significant UGG1 increase with tunicamycin or that KAR2 was highest in this condition.

Thank you for your thoughtful comment. We have better clarified the significant increase of *UGG1* expression following tunicamycin treatment and *KAR2* induction upon heat stress in the revised manuscript (Page 10, lines 216-217). Please note that Fig. 1C was revised and is now referred to as Fig. 3A.

(8) Figure 2B is not well explored/explained. There appears to be more protein in the mutant, including of higher weight in the intracellular compartment. It is difficult to ascertain if there is more too in the secretion phase with this gel. The methods do not specifically describe the concentration of protein added - just volume. Is what we are seeing a loading issue vs real differences?

Thank you for your insightful comments regarding Figure 2B. We added information on amounts of protein (30 µg per lane) in the legend of Figure 2B.

The main purpose of Fig. 2B is to examine the altered glycosylation pattern of ERQC by detecting glycoproteins using the *Galanthus nivalis* agglutinin, which specifically bind terminal α1,2-, α1,3-, and α1,6-linked mannose residues. The result of lectin blotting indicated that glycoproteins are more abundantly detected in the secretion fraction compared to in the soluble intracellular fraction, consistent with the general notion that more than 50% of secretory proteins are glycoproteins. Also, the more abundant proteins with decreased molecular weight in the secretion fraction of *ugg1*Δ mutant supported the *N*-glycan profiles with decreased hypermannosylation in *ugg1*Δ mutant. We added the purpose and more detailed interpretation on Figure 2B in the revised manuscript (Page 9, lines 174-179).

(9) Line 242 "melanin pigment" is redundant as melanin is a pigment.

We thank the reviewer for pointing out the redundancy in the phrase. We revised the text to simply state "melanin".

(10) Line 250 drops "completely" especially as the mutant did colonize the lungs of mice.

To avoid any possible misleading, we removed the term "completely" in the revised manuscript.

(11) Line 275- need to reference 18B7 as it is first introduced here.

We added the reference on the antibody 18B7 in the revised manuscript.

(12) Line 308- there are specific techniques to measure GXM size that could validate or refute the statement on "incomplete" polysaccharides. For example, DOI:10.1128/EC.00268-09.

We appreciated the valuable suggestion on specific techniques to measure GXM size, which will be one of key experiments in our future study. In the revised manuscript we cited the suggested reference to indicate the need for validation of our statement (Page 14, lines 316-318).

(13) Line 496 "mammals" - why is this used when the study is on a fungus, not a mammal? The structure of the first 2 paragraphs can be clearer to focus more on fungal biology.

We have compared both mammals and fungi to emphasize that the ERQC system is conserved among eukaryotes but diverged with a few species-specific features. This comparison is relevant in the context of understanding the evolutionary unique features of ERQC pathways in *C. neoformans*. We modified the first 2 paragraphs to clarify the main issue of our present study (Page 21, lines 472-483).

(14) Line 525- the ugg mutant was not avirulent as CFU was present and histopathology in the supplementary figures shows the tissue with ugg1 deletion was not normal (although the images are not especially easy to review). Yes, the mutant did not kill under your test conditions, but it was not avirulent (incapable of causing disease). Significantly attenuated or other descriptors should be utilized. Line 548 is also thus incorrect "complete loss of virulence".

We appreciate the reviewer’s concern regarding the description of the *ugg1*Δ mutant as avirulent. We agree that the use of merely “avirulent" may not fully capture the observed phenotypes in the CFU and histopathological data, since we cannot exclude the possibility that the *ugg1*Δ mutant retains the ability to establish an infection. Thus, we have revised the text by describing the *ugg1*Δ mutant as "almost avirulent".

(15) Line 597- the study by Fukuoka used kidney cells. It is misleading to not clearly state that this finding of ER stress was NOT done in fungi as the way it is presented makes it read as if this work was performed in C. neoformans. This should be clarified. This should also be double-checked and clarified for other statements, such as the reference to Harada in line 606, as this study used melanoma cells. These cell types are very different from cryptococcus- though I absolutely concur that lessons can be learned from comparative assessments.

We thank the reviewer for pointing out the need to clarify the experimental context of the cited studies. We explicitly stated the host cell types used in the referenced studies by Fukuoka et al. and by Harada et al., respectively, in the revised manuscript (Page 25, lines 560 and 568).